# LC-IDS: Loci-Constellation-Based Intrusion Detection for Reconfigurable Wireless Networks

**Jaime Zuniga-Mejia ***, **Rafaela Villalpando-Hernandez** , **Cesar Vargas-Rosales** and **Mahdi Zareei**

Tecnologico de Monterrey, Escuela de Ingenieria y Ciencias, Monterrey 64849, Mexico;
rafaela.villalpando@tec.mx (R.V.-H.); cvargas@tec.mx (C.V.-R.); m.zareei@tec.mx (M.Z.)
\* Correspondence: jaime.zuniga@tec.mx

**Abstract:** Detection accuracy of current machine-learning approaches to intrusion detection depends heavily on feature engineering and dimensionality-reduction techniques (e.g., variational autoencoder) applied to large datasets. For many use cases, a tradeoff between detection performance and resource requirements must be considered. In this paper, we propose Loci-Constellation-based Intrusion Detection System (LC-IDS), a general framework for network intrusion detection (detection of already known and previously unknown routing attacks) for reconfigurable wireless networks (e.g., vehicular ad hoc networks, unmanned aerial vehicle networks). We introduce the concept of 'attack-constellation', which allows us to represent all the relevant information for intrusion detection (misuse detection and anomaly detection) on a latent 2-dimensional space that arises naturally by considering the temporal structure of the input data. The attack/anomaly-detection performance of LC-IDS is analyzed through simulations in a wide range of network conditions. We show that for all the analyzed network scenarios, we can detect known attacks, with a good detection accuracy, and anomalies with low false positive rates. We show the flexibility and scalability of LC-IDS that allow us to consider a dynamic number of neighboring nodes and routing attacks in the 'attack-constellation' in a distributed fashion and with low computational requirements.

**Keywords:** distributed network intrusion detection; scalable intrusion detection; anomaly detection; misuse detection; reconfigurable networks; dimensionality reduction

## 1. Introduction

With the advent of new technologies on the horizon, such as the fifth generation of mobile communication (5G), the fourth industrial revolution, Intelligent Transportation Systems (ITS), smart cities or the Internet of Things (IoT), the number of users and range of applications for wireless communications are continuously increasing. The number of IoT connections worldwide is expected to grow from 8.6 to 22.3 billion from 2018 to 2024 [1]. As the number and range of use case scenarios for mobile communications grows, so the technical challenges associated with the network operation do. Some future applications will require massive amounts of bandwidth (e.g., virtual/augmented reality) [2]; while some other critical infrastructure applications may require Ultra-Reliable-Low-Latency-Communication (URLLC) [3], (e.g., remote surgery, vehicular communications). In order to meet the user and network demands for such a wide range of applications, wireless networks must be adaptable and reconfigurable. Reconfigurable wireless networks (RWN) [4], represent a new paradigm that allows networks to be reconfigurable at each layer of the communication stack. At the physical layer, cognitive radio techniques can be used to share spectrum between primary and secondary users, and techniques such as adaptive coding and modulation (ACM) can be used to adapt the transceivers to wireless channel phenomena (e.g., path loss, fading, interference). At the medium access control (MAC) layer, adaptive transmission rates can reduce the number of frame collisions, while transmission and sleep scheduling are necessary for energy constrained devices. At the transport

layer, congestion control techniques are usually implemented. At the network layer, decentralized topologies are considered, and the data routing must adapt to the data traffic conditions and to the dynamic network topology caused by different network phenomena, such as channel fading, or node mobility.

Because the routing process has great impact on network performance parameters (e.g., end-to-end-delay, throughput or packet delivery rate (PDR)) it is an essential mechanism to meet the user and network application demands. Different alternatives for RWN routing protocols have been proposed [5,6]. Most of these routing protocols were developed assuming a cooperative network environment, free of malicious entities. However, the open and highly dynamic nature, as well as the lack of a central organism in charge of security, make RWN vulnerable to routing attacks. A malicious node could launch a routing attack to control data traffic, to degrade network performance (e.g., sinkhole, worm hole), or to deplete network resources, such as energy or bandwidth, (e.g., flooding, rushing attack) [7,8]. For the case of critical infrastructure cyber-physical systems, network attacks may imply potential economic and human losses, thus, it is important to protect RWN from these threats [9,10].

Different secure routing protocols that rely on the encryption of the routing information have been proposed to protect the routing process in RWN [11,12]. It is worth mentioning that secure routing techniques are a necessary, but not sufficient approach for secure RWN. This is because secure routing cannot prevent all types of routing attacks, as could be the case for a selective forwarding attack, in which an attacker node discards a fraction of the packets to be forwarded, to degrade the network's throughput. Complementary techniques must be considered. Intrusion detection systems (IDS) are a set of techniques whose purpose is to identify hostile or anomalous behavior. Several IDS have been proposed to protect RWN from routing attacks [13,14]. Depending on the intrusion detection paradigm, IDS can be labeled in one of three main classes, anomaly detection, misuse detection, and hybrid approaches. Misuse-detection approaches have a good detection performance for known attacks, but are not capable of identifying unknown attacks. Anomaly-detection techniques are good to identify previously unseen threats, but cannot easily identify known attacks. Hybrid approaches are ensembles of techniques with an overall improved attack-detection performance, but with increased complexity and resource demands. It is important to mention that most proposed misuse-based and hybrid approaches in the literature, focus on a specific attack or a small set of classes of routing attacks, while anomaly-detection techniques tend to have a high false alarm rate. A more general approach that combines the complementary properties of misuse and anomaly-detection techniques is necessary to reduce the complexity of hybrid methods. Reduced complexity is essential for the implementation of IDS in low power devices (e.g., sensor node powered by energy harvesting technologies).

In this paper, we focus on the complementary detection capabilities of different IDS paradigms. Our objective is to develop a generalized mathematical framework to create an IDS capable of misuse and anomaly detection on a two-dimensional feature space, with a single distributed and lightweight intrusion detection technique. We introduce Loci-Constellation-based Intrusion Detection System (LC-IDS), which is a general lightweight and distributed technique for routing intrusion detection in RWN. The proposed approach is inspired by the root locus -based misuse-detection approach presented in recent literature [15], in which authors demonstrated the low computational workload and the attack-detection capabilities of their approach. This low computational workload is achieved by the intrinsic dimensionality-reduction capabilities of the technique. In the root locus misuse detection, each node adaptively models their neighboring nodes behavior as piecewise linear systems at a given instant. With this dynamic model, it is possible to detect routing attacks from a set of known classes of attacks, by considering the location of system poles on the $\mathcal{Z}$-plane. Then, the $\mathcal{Z}$-plane acts as an orthogonal two-dimensional feature space, which implies a reduced computational workload for the attack-detection process. This is true because malicious nodes have a dynamic behavior that is inherently

different from a regular node behavior. The authors demonstrated that their approach can be used to design individual distributed and lightweight attack detectors for a wide variety of routing attacks and network scenarios [16]. In this paper, we develop further the ideas proposed in [15], and build a framework for the attack-constellation concept. However, instead of considering the frequency domain representation of individual piecewise linear systems to design individual attack detectors, we obtain the general state-space equations that model each neighboring node. Then, we use these general state-space (SS) equations to obtain a single model that contains several attack detectors, and that has anomaly-detection capabilities. This approach allows us to extend the misuse-detection capabilities of the work presented in [15] to a more general and lightweight misuse and anomaly-detection technique. The frequency domain representation of this generalized model contains all the relevant information to represent all the known attacks on a two-dimensional feature space, the $\mathcal{Z}$-plane. Because of the fact that the $\mathcal{Z}$-plane representation of each attack considered in the general state-space model has its own root locus trajectories, we introduce the concept of 'attack-constellation', which is a visualization tool to represent all the relevant information on a two-dimensional space (similar to constellation representations of modulated signals, such as quadrature amplitude modulation (QAM)). In addition, we use this two-dimensional feature space to perform anomaly detection. This allows us to identify unknown threats, for which we can design attack detectors to include them in the general state-space model and the corresponding 'attack-constellation'. The main contributions of this paper are the following:

- We propose a general mathematical framework based on the theory of dynamical systems, to identify routing attacks and anomalous behaviors from the local perspective of an individual node in RWN. With this mathematical framework, we present LC-IDS, which is a general and distributed intrusion detection technique capable of misuse and anomaly detection.
- We introduce the concept of 'attack-constellation', which allows us to represent all the relevant information for intrusion detection on a latent 2-dimensional space. By this approach, a single node can adapt to the changing network conditions by considering a dynamic number of neighboring nodes and routing attacks to be analyzed.
- We show through simulations (including a wide range of network scenarios, including different node densities, different locations of the attack nodes, several attack severity values for the considered routing attacks and node mobility) that the proposed lightweight and distributed technique can detect already known routing attacks and previously unseen anomalies with good performance.

The rest of this paper is organized as follows, Section 2 presents background information in intrusion detection for RWN, a concise revision of relevant literature and a summary of open challenges in the state of the art, and an introduction to the root locus-based misuse detection. In Section 3, we present the definitions and notation of basic concepts used to explain our approach to intrusion detection. In Section 4, we introduce the proposed general mathematical framework for anomaly and misuse detection for routing in RWN, and we discuss the implementation of the proposed technique. Section 5 covers the experimental setup for a wide variety of network conditions simulated, we report the misuse and anomaly-detection performance rates for each case of study. Finally, in Section 6, contains the conclusions of this work.

## 2. Intrusion Detection Fundamentals and State of the Art

In this section, we present an introduction to the main issues of network intrusion detection for the routing problem in RWN, including a concise literature review. Additionally, we discuss the strengths and areas of opportunity of the main IDS paradigms and the most relevant open research challenges, which we try to overcome with our approach, to be explained in Section 4. Later we introduce the root locus-based misuse detection, which is the basis for this work.

## 2.1. Network Intrusion Detection Systems for RWN

The network intrusion detection problem consists of the identification of potentially hostile or anomalous network activities, [13,14]. In order to identify malicious activities, any IDS must perform three basic functions, data collection, intrusion detection and intrusion response. During the data collection phase, the IDS must collect and prepare relevant network metrics (e.g., application logs, information from data packets and data flows) that the intrusion detection engine will use to identify of malicious activities. The data preparation may imply techniques such as data normalization and dimensionality reduction, which are used to improve the detection performance of the intrusion detection engine. The intrusion detection engine is used to decide if there is any malicious or hostile network activity. The intrusion detection problem is, in essence, a classification problem. The intrusion detection engine must classify a given node as malicious or not malicious given the information previously collected and pre-processed by the data collection module. There are different intrusion detection methodologies for routing in RWN, the most relevant are distributed approaches, statistics-based, and machine-learning approaches. In distributed approaches, as the name implies, network nodes cooperate with each other to distribute the computational overhead of the intrusion detection task. Two popular approaches for distributing the computation of intrusion detection among the network nodes are biologically inspired techniques and trust-based techniques. Biologically inspired IDS try to create a complex global response to an attack from simple local interactions of network nodes, as in swarm intelligence-based techniques [17–19], and artificial immune systems-based approaches [20,21]. Trust-based techniques tend to have good attack-detection performance at the expense of increasing the bandwidth due to the required information exchange (e.g., trust metrics) among neighboring nodes [22–24]. Statistics-based techniques rely on statistic metrics and either static or dynamic thresholds to detect routing attacks, they tend to have accurate attack-detection performance for static and low mobility network scenarios, but for highly dynamic scenarios the obtaining of decision threshold becomes a hard challenge [25–29]. Machine learning approaches are capable of learning from the given data. For that reason, they are a good candidate for IDS in RWN because those techniques can continuously learn and adapt to the dynamic network environment and the changing network topology. Their main drawback is the high computational workload that these techniques imply to be trained and executed [30–34]. The last function of an IDS is the intrusion response, and it refers to the actions taken by the IDS after the identification of an intruder. Those actions may imply adding the potentially malicious node to a blacklist or triggering alerts to the network administrators.

The design of an IDS is a complex task, and for the particular case of IDS for routing in RWN there are some additional difficulties due to the inherent decentralized, self-organizing and dynamic nature of the network. The IDS for RWN must be implemented in nodes with severe restrictions in terms of energy, processing power, memory and bandwidth (e.g., sensor network placed at a remote location and whose nodes are powered by small batteries and energy harvesting devices). In addition, the network topology is highly dynamic due to channel fading, node mobility or sleeping schedules. This dynamic topology causes regular changes in traffic profiles, which make it difficult for modeling normal traffic behavior or the signature behavior of an attack. In the highly dynamic and stochastic nature of RWN, network performance could be affected by possible attacker nodes, and be degraded by 'natural networking' causes (e.g., node mobility, node sleep scheduling, packet loss due to traffic congestion or wireless channel impairments such as interference, multipath or fading). The lack of a central entity makes it hard to use a centralized data collection, which discards IDS with centralized architectures. The ideal IDS should be a lightweight attack-detection mechanism, capable of adapting to the rapid changes in the network conditions, robust, scalable, the time to detect any threat should be minimum to limit the damage produced by the attacker, and it would provide the necessary tools to recover from an incident.

### 2.2. Intrusion Detection Engine Paradigms

As stated previously, the intrusion detection process can be thought of as a classification problem, and there are three main paradigms for intrusion detection, anomaly detection, misuse detection, and hybrid approaches. Each paradigm has its own strengths, which we explain in this subsection and which we take advantage of, for the proposed LC-IDS, explained in detail in Section 4.

#### 2.2.1. Anomaly Detection

Anomaly detection is, in essence, an outlier detection approach, because it uses the concept of normal network state and deviations from it. This normal network state is obtained from historical records of each user's behavior. Any deviation from the obtained normal state is considered an outlier or an anomalous situation. Anomaly-detection techniques do not require prior information of the attack to be detected, this implies that there is no need for a database of known attacks. Therefore, this methodology is powerful to detect previously unseen attacks or anomalous activities, which is a useful property, given that cyber-attacks are continuously evolving. However, due to the dynamic nature of the network traffic and network topology of RWN, the normal network state could be very dynamic in time, and it could lead to a significant amount of false alarms.

#### 2.2.2. Misuse Detection

Misuse detection or signature-based detection systems usually rely on a database that stores the typical signature of all the known threats. This attack signature consists of the typical impact of the considered attack on a given set of network parameters. In order to detect a malicious node, the misuse detection system compares the behavior of each user to each attack signature in the database. Misuse detection is, in essence, a pattern-matching approach that tends to have good performance in detecting known attacks, but has difficulties in detecting unknown network anomalies. Another drawback of misuse-detection approaches is that they require a constant update of the database of known attacks, which can be a bandwidth and energy demanding process.

#### 2.2.3. Hybrid Approaches

Hybrid approaches, as the name implies, are typically ensembles of anomaly detection and misuse-detection systems. Hybrid approaches tend to have an improved attack-detection performance compared to individual approaches, this is because of the complementary detection capabilities of misuse and anomaly-detection techniques. The main drawback of hybrid approaches is the extra complexity and computational workload that those ensembles of techniques imply on the IDS. This extra complexity and computational cost may limit their range of applications to powerful host nodes. In order to implement any IDS on low power devices, it must be lightweight because of the energy and computational power constraints that some nodes could have (e.g., sensor nodes).

### 2.3. Open Challenges in the Literature

From the literature review, we can identify some of the most relevant open challenges related to IDS for routing in RWN, some of which we overcome with our proposed approach, introduced in Section 4. Those current issues can be summarized as follows:

- Network resources such as the amount of memory required, the processing workload, the used bandwidth and the time-to-detection are not commonly considered to compare the performance of different IDS. The performance evaluation for IDS is typically measured in terms of attack-detection accuracy metrics, such as the number of false negatives, the number of false positives, and detection accuracy. However, given the highly dynamic and resource constrained nature of RWN, memory, processing and bandwidth requirements are also important to the implementation of any IDS, which remain an open challenge for most of the use cases of IDS in RWN (e.g., sensor networks).

- IDS usually sacrifice attack-detection performance to reduce the resource consumption related to its implementation. Scalable, robust and lightweight IDS must be designed.
- Collaborative and hierarchical-based IDS effectively distribute the computational workload of the IDS, but they consume one of the most valuable network resources, bandwidth. This limits the scalability of some of those approaches. Hierarchical schemes help to alleviate the bandwidth consumption issue.
- The right feature space selection is crucial to achieve good attack-detection performance. However, it is difficult to find accurate normal traffic patterns or attack signatures in the complex and stochastic network environment of RWN. Machine-Learning (ML) techniques are good candidates to solve this issue, but most ML-based IDS are computationally intensive approaches which require the use of dimensionality-reduction techniques.
- A general approach for IDS is necessary to reduce the complexity of hybrid methods. The majority of proposed IDS for RWN in the literature focus on a specific attack or a small set of classes of routing attacks.
- Most of the classical machine-learning techniques do not take into consideration the temporal changes in the input data to extract useful patterns for classification.

The root locus misuse detection presented in [15], takes advantage of dynamic models that are well suited to study the time-varying nature of RWN. By this approach, it addresses some of the described open research challenges, such as the dynamism, scalability, robustness and the low demands for computational resources. The main drawback of root locus misuse detection, is that individual attack detectors must be designed for all the known attacks, and it does not have anomaly-detection capabilities to identify previously unseen threats. In the next subsection, we summarize the main ideas presented in [15], which we take as a basis to develop our general approach for intrusion detection, described in Section 4. In our approach in Section 4, we obtain the general state-space equations that model each neighboring node from the local perspective of an individual network node, instead of considering the frequency domain representation of individual piecewise linear systems to design individual attack detectors as described in [15]. Then, we use these general state-space equations to obtain a single model that contains several attack detectors and that has anomaly-detection capabilities.

### 2.4. Root Locus Based Misuse Detection

The authors in [15], proposed a mathematical framework for misuse detection for routing in RWN. This framework is based on the theory of dynamical systems, in which they consider each node as a dynamical system that models the node's dynamic behavior and its individual contribution to the network performance. The system output signals are local network performance metrics (e.g., point-to-point delay, link throughput), and the input signals are different network metrics of the channel state and the internal state of the node (e.g., signal-to-noise-ratio (SNR), number of collided frames, packets in queue). Given the dynamic nature of the network, the dynamical systems that model each node are nonlinear, but those dynamical systems can be linearized at any given instant. The authors propose a simple premise, there are inherent differences in the dynamic behavior of attacker nodes and regular nodes; and those differences will be reflected in the frequency domain representation of the models for those nodes. They propose that each node adaptively models each neighboring node's dynamic behavior as piecewise discrete-time linear systems at a given instant. Then, they use the $\mathcal{Z}$-plane representation of those models as a two-dimensional feature space for identifying neighboring malicious nodes. This reduced feature space arises naturally, independently of the number of input and output signals considered, and it does not imply any loss of information; thus, the computational workload for the attack-detection process is reduced because there is no need for additional dimensionality-reduction techniques (e.g., PCA). With this mathematical framework, the authors proposed two different IDS, the first is based on a black box system identification technique, the second IDS is based on root locus principle. For the black box approach,

the authors use a black box system identification technique to model the input–output relationship of the local performance metrics and the metrics obtained from the channel and the node state. For the root locus-based attack-detection technique, the authors take advantage of the root locus principle used in control theory, they propose some definitions of the input signals in terms of delayed metrics of the channel and node state and the relevant local performance metrics, the output signals are defined in terms of the current local performance metrics. The authors demonstrate that with those input and output signal definitions, the system poles move on predefined trajectories on the $\mathcal{Z}$-plane. They also proved that with their root locus-based approach, they could minimize the probability of classification error, by adjusting the model parameters. The authors show the intrinsic dimensionality reduction, low computational cost and good attack-detection capabilities of both techniques through a case study. They concluded that the root locus technique can be used to design individual attack detectors for an arbitrary number of routing attacks, at a lower computational cost, compared to the black box technique. For that reason, in this work, we generalize the root locus-based attack-detection approach presented in [15], to introduce a general, scalable and robust intrusion detection technique capable of misuse and anomaly detection for routing in RWN.

## 3. Basic Definitions and Notation

Let us define relevant concepts. Please note that we have modified the notation proposed in [15] because it neither allows us to represent multiple attack detectors in a single state-space model, nor it includes the anomaly-detection concept.

### 3.1. Reconfigurable Wireless Networks

RWN are dynamic entities composed of nodes connected among themselves with wireless communication links, and continuously sharing flows of information through those links in a point-to-point fashion. Those communication links can be lost or established at any moment due to different network phenomena, such as node mobility, channel fading or sleep scheduling; therefore, network topology is highly dynamic in nature.

We can describe the RWN topology at a given instant $\tau$ as a dynamic directed graph as,

$$\mathcal{G}_\tau = (\mathcal{V}_\tau, \mathcal{L}_\tau), \tag{1}$$

where

- $\mathcal{V}_\tau = \{v_i : i = 1, 2, ..., S_\tau\}$ is the set of nodes. There is a total of $S_\tau$ nodes at instant $\tau$,
- $\mathcal{L}_\tau = \{l_{ij} = (v_i, v_j) : v_i, v_j \in \mathcal{V}_\tau\}$ is the set of ordered pairs representing communication links at the same instant $\tau$. The subindices order in each link definition represents the direction of that link. If for any given order pair, $(v_i, v_j)$, the link $l_{ij} \notin \mathcal{L}_\tau$, then $\nexists$ $l_{ij}$, the link does not exists at that instant $\tau$, for any given reason (e.g., nodes are out of communication range, sleeping scheduling issues, channel fading).

Figure 1a, shows an example of a network topology $\mathcal{G}_{\tau_1}$, described at a given instant $\tau_1$; the set of nodes is $\mathcal{V}_{\tau_1} = \{v_1, v_2, v_3, v_4\}$ and the set of links is $\mathcal{L}_{\tau_1} = \{l_{12}, l_{21}, l_{13}, l_{31}, l_{34}, l_{43}\}$. Please note that links $l_{24}$, $l_{42}$ do not exist and therefore they are not in the set of links, $l_{24}, l_{42} \notin \mathcal{L}_{\tau_1}$.

### 3.2. Neighboring Nodes

Given that we are defining the mathematical framework for a distributed technique, we focus on a particular node $v_i$ and its vicinity, to explain our approach, without loss of generality. The set of neighboring nodes to a particular node $v_i \in \mathcal{V}_\tau$, at a given instant $\tau$, is defined as,

$$\mathcal{N}_i \subset \mathcal{V}_\tau = \{v_j : l_{ji} \in \mathcal{L}_\tau\}. \tag{2}$$

In Figure 1a, the set of neighboring nodes of $v_3$ is $\mathcal{N}_3 = \{v_1, v_4\}$ at instant $\tau_1$.

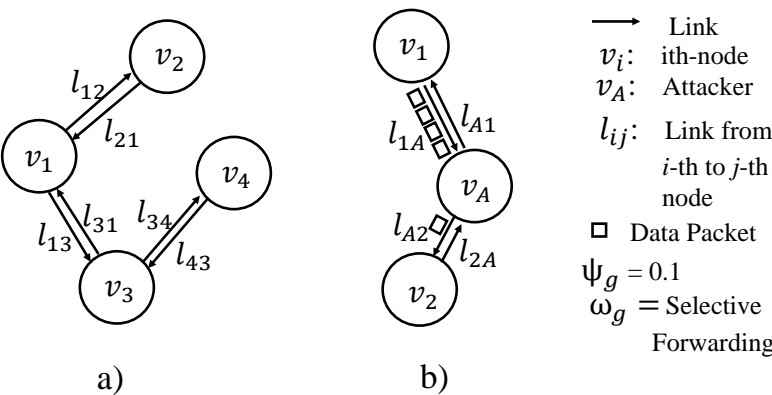

**Figure 1.** (**a**) RWN topology at a given instant $\tau_1$. (**b**) Example of a selective forwarding attack.

### 3.3. Routing Attacks

We assume that any malicious node $v_{\mathcal{A}} \in \mathcal{V}_\tau$, has access to the RWN, and that it could launch a routing attack at a given instant. The set composed of a total of $M$ known routing attacks is defined as,

$$\Omega_{\mathcal{A}} = \{\omega_g : g = 1, 2, ..., M\}. \tag{3}$$

Each $\omega_g$, has associated an attack severity metric that is bounded between a minimum and a maximum attack severity value. This attack severity metric is defined as,

$$\psi_g \in [\psi_g^{min}, \psi_g^{max}]. \tag{4}$$

As previously stated, routing attacks have an impact on network performance. The attack severity metric is defined in such a way that a greater value of attack severity corresponds to a greater impact on network performance degradation.

Figure 1b, shows an RWN in which the attacker node $v_{\mathcal{A}}$ launches the $g$-th routing attack, a selective forwarding attack for this particular example. In this selective forwarding attack, the malicious node $v_{\mathcal{A}}$, drops some of the data packets to be relayed with a probability $p_D = 0.1$. The attack severity of $\omega_g$ can be defined as the probability of a packet being dropped by the attacker node, $\psi_g = p_D$. By this attack severity definition, any increment in attack severity will correspond to a greater degradation of network performance (e.g., throughput). And being defined as a probability, the attack severity is bounded by minimum and maximum values, $\psi_g \in [\psi_g^{min} = 0, \psi_g^{max} = 1]$. In this particular example, if the attacker node $v_{\mathcal{A}}$, decides to change the attack severity, to the minimum possible value $\psi_g^{min} = 0$, this implies that not one packet will be dropped by the malicious node; and if the decision is change to the maximum possible attack severity value $\psi_g^{max} = 1$, this represents that all the data packets are discarded by $v_{\mathcal{A}}$.

Please note that given the dynamic nature of RWN and the appearance of previously unknown vulnerabilities in routing, the total number of known attacks $M$, could be varying over time. For that reason, the use of anomaly-detection-based approaches is essential to secure RWN.

### 3.4. Local Information to Detect Routing Attacks

LC-IDS uses local information to infer a global network state and to identify a potential neighboring attacker $v_{\mathcal{A}} \in \mathcal{N}_i$, launching a specific routing attack or having an anomalous behavior. We classify the local information as local performance metrics and complementary information:

- Local performance metrics. The global network performance can be thought of as a composition of individual performance contributions of each node $v_i \in \mathcal{V}_\tau$. Given that routing attacks cause network performance degradation, some local performance metrics could be used to identify hostile neighboring nodes. The concept of local performance metric refers to the performance metrics that each node $v_i \in \mathcal{V}_\tau$, can

measure from its local perspective (e.g., point-to-point delay, link throughput, link PDR). The total number of $L$ local performance metrics that a node can measure are defined in the set,

$$\mathcal{P} = \{\pi_a : a = 1, 2, .., L\}. \tag{5}$$

Each local performance metric $\pi_a$, is defined in such a way that any performance degradation corresponds to an increase in the metric. Additionally, each $g$-th routing attack $\omega_g \in \Omega_{\mathcal{A}}$ can degrade at least one local performance metric $\pi_a \in \mathcal{P}$ and each $\pi_a \in \mathcal{P}$ can be affected by one or more attacks. Please note that the subindex in each $\pi_a \in \mathcal{P}$ is used to identify each metric in the set, but it does not indicate any particular order.

The subset $\mathcal{P}_g$, composed of a total of $\lambda_p \leq L$, of local performance metrics that the misuse detection part of LC-IDS uses to detect the $g$-th routing attack is defined as,

$$\mathcal{P}_g = \{\pi_a \in \mathcal{P} : \omega_g \text{ can be detected}\}, |\mathcal{P}_g| = \lambda_p. \tag{6}$$

- Complementary information. Given that routing attacks are one, but not the only possible cause of network performance degradation, we need to consider complementary information that allows us to discriminate routing attacks. The set of complementary network metrics that a node $v_i$ can measure to discard routing attacks is defined as,

$$X = X_{\mathcal{A}} \cup X_{\mathcal{N}}, \tag{7}$$

where

- $X_{\mathcal{A}} = \{\chi_{\mathcal{A}_b} : b = 1, 2, ..., A\}$, is the set of local network metrics that are related to routing attacks and performance degradation (e.g., a large number of routing control messages may indicate a possible RREQ flooding attack), there are a total of $A$ elements in the set, $|X_{\mathcal{A}}| = A$,
- $X_{\mathcal{N}} = \{\chi_{\mathcal{N}_c} : c = 1, 2, ..., N\}$ is the set of local network metrics related to 'natural' performance degradation (e.g., a low link throughput may be caused by channel congestion, measured with the number of colliding frames per time unit, or by the number of packets discarded in queue per time unit). There is a total of $N$ elements in the set, $|X_{\mathcal{N}}| = N$.

The subindices $b$ and $c$ in $\chi_{\mathcal{A}_b} \in X_{\mathcal{A}}$ and $\chi_{\mathcal{N}_c} \in X_{\mathcal{N}}$, help to identify each metric in their respective set, those subindices do not indicate any order in the metrics.

Because a lightweight intrusion detection technique is essential for its implementation on low power devices, we do not need to consider every network metric $\chi_{\mathcal{A}_b} \in X_{\mathcal{A}}$, $\chi_{\mathcal{N}_c} \in X_{\mathcal{N}}$, to detect the $g$-th routing attack; but we can consider subsets of relevant metrics to identify each routing attack $\omega_g$. Those subsets $X_{\mathcal{A}g} \subset X_{\mathcal{A}}$, and $X_{\mathcal{N}g} \subset X_{\mathcal{N}}$, are defined as, $X_{\mathcal{A}g} = \{\chi_{\mathcal{A}_b} \in X_{\mathcal{A}} : \omega_g \text{ can be detected}\}$, whose cardinality is $|X_{\mathcal{A}g}| = \lambda_{ag}$, and, $X_{\mathcal{N}g} = \{\chi_{\mathcal{N}c} \in X_{\mathcal{A}} : \omega_g \text{ can be detected}\}$, whose cardinality is $|X_{\mathcal{N}g}| = \lambda_{ng}$.

Please note that the reduced cardinality of the subsets of complementary network metrics, contribute to a lower computational workload of LC-IDS; $|X_{\mathcal{A}g}| = \lambda_{ag} < |X_{\mathcal{A}}| = A$; $|X_{\mathcal{N}g}| = \lambda_{ng} < |X_{\mathcal{N}}| = N$.

### 3.5. LC-IDS Architecture

The authors in [15], take a divide and conquer strategy, in which each node $v_i$ obtains an adaptive Linear Shift-Invariant (LSI) system model $IDS_{ij,\omega_g}$, for each neighboring node $v_j \in \mathcal{N}_i$ and for each routing attack $\omega_g \in \Omega_{\mathcal{A}}$, every time period $kT, k = 0, 1, 2, ....$ An LSI system is a mathematical model that describes the dynamical relation between the discrete input and output signals of the system of interest (e.g., network node). By modeling the dynamic behavior of network nodes at a given instant, we can identify malicious nodes because of their inherently different dynamic behavior from the rest of the network nodes. Figure 2a, shows this approach, implemented in a node $v_i$ with at least one

neighboring node, $\mathcal{N}_i = \{v_1, ...\}$. The IDS implemented in $v_i$ can be decomposed in $IDS_{i1}$, $IDS_{i2}$, ... Each $IDS_{ij}$, can be further decomposed in attack detectors for each routing attack in $\Omega_{\mathcal{A}} = \{\omega_1, ..., \omega_M\}$. This approach allows them to design individual robust and lightweight misuse detectors for an arbitrary number of neighboring nodes and classes of routing attacks; however, each $IDS_{ij}$ is independent of each other, which implies that anomaly detection cannot be performed directly, without considering additional techniques. In Figure 2b, we show the architecture of the proposed LC-IDS, in which each node adaptively models the dynamic behavior of its neighboring nodes; only one LC-IDS is modeled for each neighboring node $v_j \in \mathcal{N}_i$. Each $LC - IDS_{ij}$, can perform anomaly detection and misuse detection of all the known routing attacks $\omega_g \in \Omega_{\mathcal{A}}$. In Section 4.2 we describe the dynamic behavior of the LSI that models each $LC - IDS_{ij}$, in Section 4.4 and Appendix A we explain the theoretical framework that allows us to perform misuse and anomaly detection in the same lightweight IDS. This theoretical framework for general intrusion detection is the main contribution of this work.

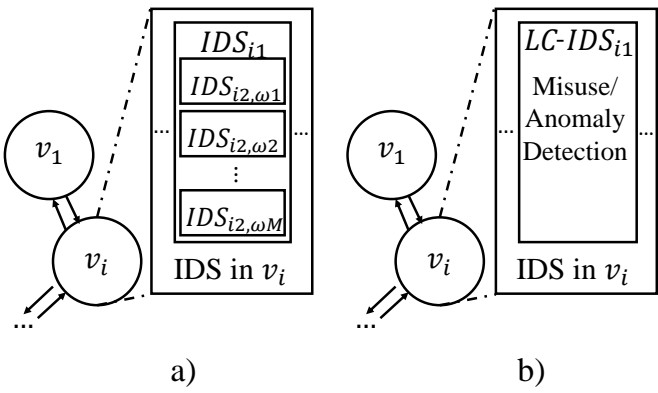

a)                                          b)

**Figure 2.** (**a**) Root locus-based misuse detection. (**b**) LC-IDS anomaly and misuse-detection architecture.

Table 1 summarizes and defines the notation of the basic concepts presented in this section.

**Table 1.** Summary of concepts and notation.

| Notation | Description |
|---|---|
| $\mathcal{G}_\tau$ | Network topology graph at instant $\tau$ |
| $\mathcal{V}_\tau$ | Set of nodes at instant $\tau$ |
| $v_i$ | $i$-th node |
| $\mathcal{L}_\tau$ | Set of links at instant $\tau$ |
| $l_{ij}$ | Link that goes from the $i$-th node to the $j$-th node |
| $\mathcal{N}_i$ | Set of neighboring nodes of the $i$-th node |
| $\Omega_{\mathcal{A}}$ | Set of known routing attacks |
| $\omega_g$ | $g$-th known routing attack |
| $\psi_g$ | Attack severity metric of the $g$-th routing attack |
| $\mathcal{P}_g$ | Set of local performance metrics degraded by $\omega_g$ |
| $\pi_a$ | $a$-th performance metric |
| $X_{\mathcal{A}}$ | Set of local metrics related to $\omega_g$ |
| $\chi_{\mathcal{A}_b}$ | $a$-th local metric related to $\omega_g$ |
| $X_{\mathcal{N}}$ | Set of local metrics not related to $\omega_g$ |
| $\chi_{\mathcal{N}_c}$ | $c$-th local metric not related to $\omega_g$ |
| $a(z)$ | Polynomial in the $\mathcal{Z}$-plane |
| $a(k)$ | Time series |
| $a$ | Scalar |
| $\boldsymbol{a}$ | Vector/matrix |
| $a^{\mathsf{T}}$ | Transpose operator |
| $a^{-1}$ | Inverse operator |
| $|a|$ | Modulus operator |
| $||a||_2$ | Euclidean norm operator |

## 4. Loci-Constellation-Based Intrusion Detection System (LC-IDS)

In this section, we define the general mathematical framework for intrusion detection engine of our proposed technique, which uses the local data collection approach described in Section 3.4. We discuss the implementation of online attack and anomaly detection engine of LC-IDS. Online attack detection is performed in real time, unlike forensics approaches, in which network data are analyzed after the network attack. Please note that local data collection allows the method to operate without scarifying the network resources such as bandwidth, synchronization and node power battery, which result very convenient in dense sensor networks. Our objective is to develop a generalized mathematical framework to create an IDS capable of misuse and anomaly detection on a two-dimensional feature space, with a single distributed and lightweight intrusion detection technique. We will take advantage of the state-space representation of the dynamic behavior of neighboring nodes to detect known routing attacks and previously unseen network anomalies.

### 4.1. Parametric Autoregressive Model

We begin the definition of the mathematical framework for the proposed intrusion detection technique by the misuse detection part of LC-IDS. Our objective is to obtain an adaptive LSI system that models the dynamic behavior of each neighboring node $v_j \in \mathcal{N}_i$, to later use the $\mathcal{Z}$-plane representation of that model as a two-dimensional feature space to detect each known attack $\omega_g \in \Omega_{\mathcal{A}}$. Then, we define the methodology for anomaly detection in the obtained feature space.

Without loss of generality, and in order to develop the LSI model for $LC - IDS_{ij}$, we focus on the $g$-th routing attack and we consider each neighboring node $v_j \in \mathcal{N}_i$ as an LSI system, which has been linearized for a small time-window around a given instant, $\tau$. This approach can later be used for all the routing attacks $\omega_g \in \Omega_{\mathcal{A}}$. We take periodic samples, with a sampling period $T$, of the relevant local network metrics $\pi_a$, $\chi_{\mathcal{A}_b}$ and $\chi_{\mathcal{N}_c}$, to form the time series, $\pi_a(k)$, $\chi_{\mathcal{A}_b}(k)$ and $\chi_{\mathcal{N}_c}(k)$; $k = 0, 1, 2, ....$ We propose the multivariate linear regression to model the relationship of the time series in the time domain as,

$$\pi_a(k) = \sum_{b=1}^{A} \alpha_b \chi_{\mathcal{A}_b}(k) 1_{X_{\mathcal{A}_g}}(\chi_{\mathcal{A}_b}) + \sum_{c=1}^{N} \beta_c \chi_{\mathcal{N}_c}(k) 1_{X_{\mathcal{N}_g}}(\chi_{\mathcal{N}_c}) + \gamma_g(k), \tag{8}$$

where $\alpha_b, \forall b, \beta_c, \forall c, \gamma_g(k)$, are the model parameters to be estimated each period, $1_{X_{\mathcal{A}_g}}(\chi_{\mathcal{A}_b})$ and $1_{X_{\mathcal{N}_g}}(\chi_{\mathcal{N}_c})$ are indicator functions, defined as,

$$1_{X_{\mathcal{A}_g}}(\chi_{\mathcal{A}_b}) = \begin{cases} 0 & \text{if } \chi_{\mathcal{A}_b} \notin X_{\mathcal{A}_g} \\ 1 & \text{if } \chi_{\mathcal{A}_b} \in X_{\mathcal{A}_g} \end{cases}, \tag{9}$$

and,

$$1_{X_{\mathcal{N}_g}}(\chi_{\mathcal{N}_c}) = \begin{cases} 0 & \text{if } \chi_{\mathcal{N}_c} \notin X_{\mathcal{N}_c} \\ 1 & \text{if } \chi_{\mathcal{N}_c} \in X_{\mathcal{N}_c} \end{cases}. \tag{10}$$

Please note that if the $\chi_{\mathcal{A}b} \notin X_{\mathcal{A}g}$, the corresponding term in the summation is zero and has no effect on $\sum_{b=1}^{A} \alpha_b \chi_{\mathcal{A}_b}(k) 1_{X_{\mathcal{A}_g}}(\chi_{\mathcal{A}_b})$. A similar argument can be made for each $\chi_{\mathcal{N}_c} \notin X_{\mathcal{N}g}$ and $\sum_{c=1}^{N} \beta_c \chi_{\mathcal{N}_c}(k) 1_{X_{\mathcal{N}_g}}(\chi_{\mathcal{N}_c})$. The use of indicator functions allows us to obtain the general state-space representation of the dynamical system, as shown in Section 4.4 and Appendix A.

For each routing attack, we select one local performance metric $\pi_a \in \mathcal{P}_g$, each $\chi_{\mathcal{A}_b} \in X_{\mathcal{A}g}$, and each $\chi_{\mathcal{N}_c} \in X_{\mathcal{N}g}$. Because $|X_{\mathcal{A}_g}| = \lambda_{ag}$ and $|X_{\mathcal{N}g}| = \lambda_{ng}$; the multivariate regression model for the $g$-th routing attack, has a total number of $\lambda_{ag} + \lambda_{ng} + 1$ parameters. In Equation (8), we make the distinction between the time series of network metrics sensitive to routing attacks, $\chi_{\mathcal{A}_b}(k)$, and the time series of network metrics non-sensitive to attacks, $\chi_{\mathcal{N}_c}(k)$. This distinction allows each attack detector of $LC - IDS_{ij}$ to be sensitive

to each routing attack $\omega_g \in \Omega_{\mathcal{A}}$, and not to other factors that may cause performance degradation (e.g., channel fading), of the $a$-th local performance metric $\pi_a \in \mathcal{P}_g$. It is worth mentioning that in Equation (8), we do not consider the relationship between delayed output and input signal and the current output signal, which is an essential characteristic of dynamical systems; but, the dynamical behavior of $LC - IDS_{ij}$ is originated from the input and output signals, which will be defined in Section 4.2.

For the model in Equation (8), we obtain a set of parameters $\{\alpha_b : b = 1, ..., \lambda_a\}$, which relate the time series of the performance metric $\pi_a(k)$, and the time series of the $b$-th metric sensitive to routing attacks, $\chi_{\mathcal{A}_b}(k)$. Similarly, the set of parameters $\{\beta_c : c = 1, ..., \lambda_n\}$, represent the relationship between the time series of the network metrics non-sensitive to routing attacks $\chi_{\mathcal{N}_c}(k)$ and the local performance metric $\pi_a(k)$; $\gamma_g(k)$ is a free parameter, whose value is fully determined by the data, at each sampling period. The model parameters can be obtained by linear regression, considering a number $d$, of delayed measurements of the time series, in a sliding-time-window fashion. A longer time-window length $d$ may capture longer time trends in the data, at the expense of more computational workload.

### 4.2. Desired Dynamic Response and 'Attack-Constellation'

Let us define the concept of 'attack-constellation', as the two-dimensional feature space, in which we can represent all the relevant information to perform anomaly detection, and misuse detection of the known routing attacks $\omega_g \in \Omega_{\mathcal{A}}$. At a given instant, the system poles of the $LC - IDS_{ij}$, can be represented in this 'attack-constellation'; and depending on their location, the node $v_i$ can decide if the $j$-th neighboring node is an attacker. Every attack detector for each known attack $\omega_g \in \Omega_{\mathcal{A}}$, has its corresponding pair of system poles. Given that at any particular instant, there are a total number of $M$ known routing attacks, and because complex poles have a conjugate pair, the system representation of $LC - IDS_{ij}$ is of order $2M$. These poles, by definition, tend to be near the origin of the $\mathcal{Z}$-plane in absence of the attack $\omega_g \in \Omega_{\mathcal{A}}$; $z_{\mathcal{N}g}^{min} = \bar{z}_{\mathcal{N}g}^{min} = 0$. In addition, in the presence of an attack $\omega_g \in \Omega_{\mathcal{A}}$, one of the two system poles for that attack detector moves closer to an arbitrary location, $z_{\mathcal{A}g}^{max} = r_g \cos\theta_g + jr_g \sin\theta_g$; the conjugate pole moves to $\bar{z}_{\mathcal{A}g}^{max} = r_g \cos\theta_g - jr_g \sin\theta_g$. Therefore, we obtain a region on the $\mathcal{Z}$-plane that represents the absence of the corresponding $g$-th routing attack. This region is common for all the misuse detectors in $LC - IDS_{ij}$, and is located near the origin of the $\mathcal{Z}$-plane. As an example, consider Figure 3, which shows a total of four known routing attacks, $\Omega_{\mathcal{A}} = \{\omega_1, \omega_2, \omega_3, \omega_4\}$. Please note that a given instant, we can represent the $2M$ system poles, and depending on how far they are from the origin, we can assign a probability to identify a potential malicious neighboring node $v_j \in \mathcal{N}_i$. Later in this section, we propose a methodology to define the decision boundary for the non-attack region.

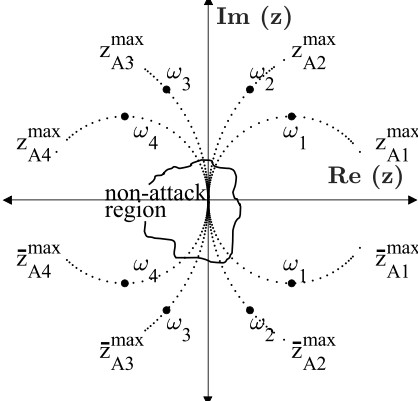

**Figure 3.** 'Attack-constellation', representing the system poles at a given instant $\tau$. Each pole in the constellation is sensitive to a specific attack, $\omega_g \in \Omega_{\mathcal{A}}$. The further the poles from the origin, the greater the value of the probability of a given routing attack $\omega_g$.

The LSI system that models the dynamic behavior of the *j*-th neighboring node at a given instant, is of order $2M$; thus, the characteristic equation of the dynamic model of $LC - IDS_{ij}$ can be stated as a $2M$ degree polynomial $Q(z)$,

$$Q(z) = \prod_{g=1}^{M} Q_g(z);$$
(11)

where each $Q_g(z)$ is a second-degree polynomial, given by,

$$
\begin{aligned}
Q_g(z) &= 1 + \left( \sum_{b=1}^{A} \alpha_b^{\eta_g} 1_{X_{A_g}}(\chi_{A_b}) \right) \frac{(z^2 - 2zr_g \cos\theta_g + r_g^2)}{z^2} \\
&= 1 + \sum_{b=1}^{A} \alpha_b^{\eta_g} 1_{X_{A_g}}(\chi_{A_b}) - z^{-1} \left( 2r_g \cos\theta_g \sum_{b=1}^{A} \alpha_b^{\eta_g} 1_{X_{A_g}}(\chi_{A_b}) \right) \\
&\quad + z^{-2} \left( r_g^2 \sum_{b=1}^{A} \alpha_b^{\eta_g} 1_{X_{A_g}}(\chi_{A_b}) \right).
\end{aligned}
$$
(12)

Please note that each $Q_g(z)$, is defined as the characteristic equation of a closed-loop system, whose poles go from $z_{\mathcal{N}_g}^{min} = \bar{z}_{\mathcal{N}_g}^{min} = 0$, to $z_{A_g}^{max}$ and $\bar{z}_{A_g}^{max}$, as the value of $\sum_{b=1}^{A} \alpha_b^{\eta_g} 1_{X_{A_g}}(\chi_{A_b})$, increases from zero to infinity. Each polynomial $Q_g(z)$ has three parameters, $r_g$, $\eta_g$ and $\theta_g$. Each parameter $\theta_g$ is chosen arbitrarily for each routing attack, this $\theta_g$ defines the trajectories that the pair of poles of $Q_g(z)$ will follow. The values of the parameters $r_g$ and $\eta_g$ must be found in such a way that optimize the detection performance for the $g$-th routing attack detector. Later in this section, we propose a methodology to find these optimal parameters values.

### 4.3. Input and Output Signals

As previously stated, the multivariate linear regression model in Equation (8), does not capture the desired dynamical behavior of the system, whose poles on the $\mathcal{Z}$-plane move on the trajectories defined by the 'attack-constellation' diagram in Figure 3. That desired dynamical behavior of the system comes from the input signals $u_{A_b}(k)$ and $u_{\mathcal{N}_c}(k)$, and output signal $y_g(k)$, defined as,

$$u_{A_b}(k) = \chi_{A_b}(k) + \alpha_b^{\eta_g - 1} y_a(k) - 2r_g \cos\theta_g \alpha_b^{\eta_g - 1} y_a(k-1) + r_g^2 \alpha_b^{\eta_g - 1} y_a(k-2),$$
(13)

$$u_{\mathcal{N}_c}(k) = \chi_{\mathcal{N}_c}(k),$$
(14)

$$y_g(k) = \pi_a(k) - \gamma_g(k),$$
(15)

where $a = 1, ..., \lambda_{pg}$, $b = 1, ..., \lambda_{ag}$ and $c = 1, ..., \lambda_{ng}$. Derivation of the input and output signals that lead to the desired dynamic response of an individual attack-detection model can be found in the Appendix in [15]. The analysis in the following sections is different from that presented in [15].

### 4.4. State-Space Representation

In this subsection, we present the state-space representation of the LSI system that models the dynamic behavior of $LC - IDS_{ij}$. This state-space representation is obtained from the parametric autoregressive models introduced in Section 4.1, and the input–output signal definitions in Section 4.3. There is one autoregressive model, and two system poles, for each routing attack $\omega_g \in \Omega_A$, which lead to the LSI system of order $2M$ described in Section 4.2. The derivation of the proposed state-space representation can be found in Appendix A.

The state transition equation is given by,

$$x(k+1) = A(k)x(k) + B(k)u(k), \tag{16}$$

where $x(k+1)$ represents the state vector at the next time period, $x(k)$ is the current state vector, $u(k)$ is the input signal vector, $A(k)$ is the state matrix, and $B(k)$ is the input-to-state matrix.

The state transition model in (16) in matrix form is,

$$
\underbrace{\begin{bmatrix} x_1^{(1)}(k+1) \\ x_1^{(2)}(k+1) \\ x_2^{(1)}(k+1) \\ x_2^{(2)}(k+1) \\ \vdots \\ x_M^{(1)}(k+1) \\ x_M^{(2)}(k+1) \end{bmatrix}}_{(2M\times 1)} = \underbrace{\begin{bmatrix} \underset{(2\times2)}{A_1(k)} & \underset{(2\times2)}{0} & \cdots & \underset{(2\times2)}{0} \\ \underset{(2\times2)}{0} & \underset{(2\times2)}{A_2(k)} & \cdots & \underset{(2\times2)}{0} \\ \vdots & \vdots & \ddots & \vdots \\ \underset{(2\times2)}{0} & \underset{(2\times2)}{0} & \cdots & \underset{(2\times2)}{A_M(k)} \end{bmatrix}}_{(2M\times 2M)} \underbrace{\begin{bmatrix} x_1^{(1)}(k) \\ x_1^{(2)}(k) \\ x_2^{(1)}(k) \\ x_2^{(2)}(k) \\ \vdots \\ x_M^{(1)}(k) \\ x_M^{(2)}(k) \end{bmatrix}}_{(2M\times 1)}
$$

$$
+ \underbrace{\begin{bmatrix} \underset{(2\times[A+N])}{B_1(k)} \\ \vdots \\ \underset{(2\times[A+N])}{B_M(k)} \end{bmatrix}}_{(2M\times[A+N])} \underbrace{\begin{bmatrix} u_{\mathcal{A}_1}(k) \\ \vdots \\ u_{\mathcal{A}_A}(k) \\ u_{\mathcal{N}_1}(k) \\ \vdots \\ u_{\mathcal{N}_N}(k) \end{bmatrix}}_{([A+N]\times 1)}, \tag{17}
$$

where each state variable has a subindex that relates it to a given attack $\omega_g$; similarly, the superindex in each state variable denotes the time period from which that state variable was derived.

Each submatrix $A_g(k)$ in (17), is defined as,

$$
A_g(k) = \underbrace{\begin{bmatrix} \dfrac{2r_g \cos\theta_g \sum_{b=1}^{A} \alpha_b^{\eta_g} 1_{X_{\mathcal{A}_g}}(\chi_{\mathcal{A}_b})}{1+\sum_{b=1}^{A} \alpha_b^{\eta_g} 1_{X_{\mathcal{A}_g}}(\chi_{\mathcal{A}_b})} & 1 \\ \dfrac{-r_g^2 \sum_{b=1}^{A} \alpha_b^{\eta_g} 1_{X_{\mathcal{A}_g}}(\chi_{\mathcal{A}_b})}{1+\sum_{b=1}^{A} \alpha_b^{\eta_g} 1_{X_{\mathcal{A}_g}}(\chi_{\mathcal{A}_b})} & 0 \end{bmatrix}}_{(2\times2)}, \tag{18}
$$

and each submatrix $B_g(k)$ is defined as,

$$
B_g(k) = \underbrace{\begin{bmatrix} 0 & \cdots & 0 \\ \dfrac{\alpha_1 1_{X_{\mathcal{A}_g}}(\chi_{\mathcal{A}_1})}{1+\sum_{b=1}^{A} \alpha_b^{\eta_g} 1_{X_{\mathcal{A}_g}}(\chi_{\mathcal{A}_b})} & \cdots & \dfrac{\beta_N 1_{X_{\mathcal{N}_g}}(\chi_{\mathcal{N}_N})}{1+\sum_{b=1}^{A} \alpha_b^{\eta_g} 1_{X_{\mathcal{A}_g}}(\chi_{\mathcal{A}_b})} \end{bmatrix}}_{(2\times[A+N])}. \tag{19}
$$

Please note that the matrices $A(k)$ and $B(k)$ in (16), are time-varying, because each submatrix $A_g(k)$ and $B_g(k)$ depend on the last estimated values of the multivariate linear model parameters $\alpha_b$, and $\beta_c$, from Equation (8).

The output equation is,

$$y(k) = Cx(k), \tag{20}$$

$$\underbrace{\begin{bmatrix} y_1(k) \\ \hline y_2(k) \\ \hline \vdots \\ \hline y_M(k) \end{bmatrix}}_{(M \times 1)} = \underbrace{\begin{bmatrix} 1 & 0 & 0 & \dots & 0 & 0 \\ \hline 0 & 0 & 1 & \dots & 0 & 0 \\ \hline \vdots & \vdots & \vdots & \dots & \vdots & \vdots \\ \hline 0 & 0 & 0 & \dots & 1 & 0 \end{bmatrix}}_{(M \times 2M)} \underbrace{\begin{bmatrix} x_1^{(1)}(k) \\ x_1^{(2)}(k) \\ \hline x_2^{(1)}(k) \\ x_2^{(2)}(k) \\ \hline \vdots \\ \hline x_M^{(1)}(k) \\ x_M^{(2)}(k) \end{bmatrix}}_{(2M \times 1)}, \tag{21}$$

where $\boldsymbol{y}(k)$ is the output signal vector and $\boldsymbol{C}$ is the state-to-output matrix.

The system poles $z_{ij}$, which model the dynamical behavior of $LC - IDS_{ij}$, are obtained from the state-space representation as,

$$z_{ij} = z_{ij} : |z_{ij}\boldsymbol{I} - \boldsymbol{A}(k)| = 0; \tag{22}$$

to then, be used as features by $LC - IDS_{ij}$ to detect anomalous network behavior or each known routing attack $\omega_g \in \Omega_{\mathcal{A}}$.

### 4.5. Misuse-Detection Decision Rule

The misuse detection part of $LC - IDS_{ij}$, performs a classification task, in which it has to assign the current neighbor $v_j$ to a class from the set $\{\mathcal{C}_{\mathcal{A}g}, \mathcal{C}_{\mathcal{N}g}\}$; where $\mathcal{C}_{\mathcal{A}g}$, corresponds to the class in which the $j$-th neighboring node is identified as an $\omega_g$-attacker; and $\mathcal{C}_{\mathcal{N}g}$, is the class that corresponds to the non-$\omega_g$-attacker nodes. This classification is made by considering the system poles $z_{ij}$, in the reduced feature space of the $\mathcal{Z}$-plane, and a decision rule $h_g(|z|)$, for each attack detector, $g = 1, 2, ..., M$.

Recall that the optimal values for the parameters $r_g$ and $\eta_g$ have not been defined for any $Q_g(z)$. Given concrete values for $r_g$ and $\eta_g$, we define a probability density function (pdf) for the modulus of the pole locations when the network is not under the $g$-th routing attack, $|z_{\mathcal{N}g}| = |\bar{z}_{\mathcal{N}g}|$, as $f_{\mathcal{N}g}(|z_{\mathcal{N}g}|)$. Similarly, we define the pdf for the modulus of the pole locations, $|z_{\mathcal{A}g}| = |\bar{z}_{\mathcal{A}g}|$, when there is an attack $\omega_g$ (with severity $\psi_g$), as $f_{\mathcal{A}g}(|z_{\mathcal{A}g}|)$. Then, we use decision theory to define each decision rule $h_g(|z|)$ that allows $LC - IDS_{ij}$ to detect the $g$-th routing attack $\omega_g$. Since we define the pdf of the pole clusters as a function of the modulus of the poles, the decision rule $h_g(|z|)$ can be defined with the decision threshold $th_g$. Thus, the decision rule $h_g(|z|)$, can be expressed as,

$$h_g(|z|) = \mathcal{C}_{\mathcal{A}g}, \tag{23}$$

if and only if

$$|z| > th_g, \tag{24}$$

where the decision threshold $th_g$, is evaluated at the pole $z$ associated with the constellation branch of $\omega_g$, and it is defined as,

$$th_g = z : f_{\mathcal{A}g}(|z_{\mathcal{A}g}|)|_z = f_{\mathcal{N}g}(|z_{\mathcal{N}g}|)|_z. \tag{25}$$

Let $P_g(\epsilon)$ be the probability of $LC - IDS_{ij}$ making a classification mistake. Please note that the decision threshold $th_g$, and the probability of error $P_g(\epsilon)$, depend on the selected values of the parameters $r_g$ and $\eta_g$, for the corresponding attack detector. Thus, we state $P_g(\epsilon)$ as a function of $r_g$ and $\eta_g$, $P_g(\epsilon) = f_g(r_g, \eta_g)$. Then, we define some constraints as follows; the expected values of the poles modulus in absence of attack are restricted by a small value $\zeta_g \approx 0$, $E[|z_{\mathcal{N}g}|] \leq \zeta_g$. Another restriction is that the modulus of the expected value of the poles during an attack condition must be greater than $\zeta_g$ and smaller than an

arbitrary value $\xi_g$; $\zeta_g < E[|z_{\mathcal{A}g}|] < \xi_g$. Finally, we select the values $r_g$ and $\eta_g$ that minimize probability of error $P_g(\epsilon)$, i.e.,

$$
\begin{aligned}
\underset{r_g, \eta_g}{\text{minimize}} \; & P_g(\epsilon) = f_g(r_g, \eta_g), \\
\text{subject to} & \\
& E[|z_{\mathcal{N}g}|] \leq \zeta_g, \\
& \zeta_g < E[|z_{\mathcal{A}g}|] < \xi_g.
\end{aligned}
\tag{26}
$$

### 4.6. Anomaly Detection

As previously stated, the 'attack-constellation' contains a pair of system poles that tend to move away from the origin on the $\mathcal{Z}$-plane, as the attack severity metric $\psi_g$, increases for each corresponding routing attack $\omega_g \in \Omega_{\mathcal{A}}$. Let's assume that we know the probability distribution for the poles in the absence of the $g$-th attack $f_{\mathcal{N}g}(|z_{\mathcal{N}g}|)$, for each known routing attack $\omega_g \in \Omega_{\mathcal{A}}$ considered in the 'attack-constellation'. Then, we can obtain the mean $\mu_g$, and standard deviation $\sigma_g$, for each $f_{\mathcal{N}g}(|z_{\mathcal{N}g}|)$, to form the column vector $\boldsymbol{\Phi} \in \mathcal{R}^{2M}$, given by,

$$
\boldsymbol{\Phi} = [\mu_1, \sigma_1, \ldots, \mu_M, \sigma_M]^{\mathsf{T}}.
\tag{27}
$$

This vector $\boldsymbol{\Phi}$ contains relevant statistical information about the non-anomalous dynamic behavior of a neighboring node $v_j \in \mathcal{N}_i$. The statistical information in $\boldsymbol{\Phi}$ can be obtained from a large set of historic non-anomalous data. Then, we can use this non-anomalous vector $\boldsymbol{\Phi}$ as a reference to perform online outlier detection. Consider that we obtain the mean $\mu_{g, d_{\boldsymbol{\phi}}}$, and standard deviation $\sigma_{g, d_{\boldsymbol{\phi}}}$, of the moduli of the 'attack-constellation' poles; those statistic metrics are obtained from a small time-window $d_{\boldsymbol{\phi}}$ that includes the current and previous system poles in the 'attack-constellation'. Then, with those statistical values, we define the vector $\boldsymbol{\phi} \in \mathcal{R}^{2M}$, as,

$$
\boldsymbol{\phi} = [\mu_{1, d_{\boldsymbol{\phi}}}, \sigma_{1, d_{\boldsymbol{\phi}}} \ldots, \mu_{M, d_{\boldsymbol{\phi}}}, \sigma_{M, d_{\boldsymbol{\phi}}}]^{\mathsf{T}}.
\tag{28}
$$

Please note that the vector $\boldsymbol{\phi}$ contains relevant temporal and statistical information about the dynamical behavior of the neighboring node $v_j \in \mathcal{N}_i$. Therefore, for non-anomalous data, vector $\boldsymbol{\phi}$ must be similar to the reference vector $\boldsymbol{\Phi}$. We can use the Euclidean distance $s_{\boldsymbol{\Phi}}$, as a measure of similarity between $\boldsymbol{\phi}$ and $\boldsymbol{\Phi}$, as,

$$
s_{\boldsymbol{\Phi}} = ||\boldsymbol{\Phi} - \boldsymbol{\phi}||_2 = \sqrt{(\mu_1 - \mu_{1, d_{\boldsymbol{\phi}}})^2 + \ldots + (\sigma_M - \sigma_{M, d_{\boldsymbol{\phi}}})^2}.
\tag{29}
$$

With historical non-anomalous data, we can obtain an empirical cumulative probability distribution (cdf) $F_{s_{\boldsymbol{\phi}}}(s_{\boldsymbol{\phi}})$ for non-anomalous data. With this cdf, we can obtain a decision rule $h_{\boldsymbol{\Phi}}$, to decide if the current dynamic behavior of the $j$-th neighboring node $v_j \in \mathcal{N}_i$ is anomalous, and that neighboring node belongs to the subset of neighboring nodes with anomalous behavior $\mathcal{N}_{\mathcal{A}} \subset \mathcal{N}_i$. The decision rule for the anomaly-detection engine of $LC - IDS_{ij}$ is given by,

$$
h_{\boldsymbol{\Phi}} = \mathcal{N}_{\mathcal{A}}
\tag{30}
$$

if and only if

$$
s_{\boldsymbol{\Phi}} > th_{\boldsymbol{\Phi}},
\tag{31}
$$

where the decision threshold $th_{\boldsymbol{\phi}}$, is defined as,

$$
th_{\boldsymbol{\Phi}} = s_{\boldsymbol{\Phi}} : F_{s_{\boldsymbol{\Phi}}}(s_{\boldsymbol{\Phi}}) = p_{\boldsymbol{\Phi}};
\tag{32}
$$

$p_{\boldsymbol{\Phi}} \in [0, 1]$, is a design parameter that represents a probability value of the non-anomalous data instances correctly classified as non-anomalous. The value of $p_{\boldsymbol{\Phi}}$ must be selected to be close to one because it represents the non-anomalous detection accuracy, thus, the closer the value of $p_{\boldsymbol{\Phi}}$ to one, the lower the number of false positives in anomaly detection.

Figure 4a, shows an example of an 'attack-constellation' of one known attack, the reference vector $\mathbf{\Phi} \in \mathcal{R}^2$, and several instances of the vector $\boldsymbol{\phi}$. In Figure 4b, we present the empirical cdf for the distance metric $s_\mathbf{\Phi}$, from the example in Figure 4a, and the corresponding decision threshold $th_\mathbf{\Phi}$.

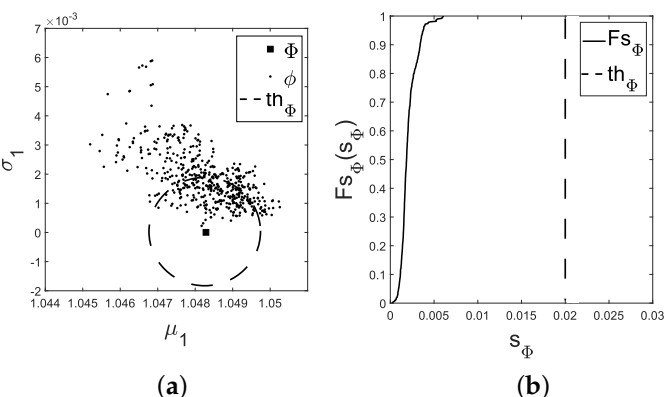

|  (a)  |  (b)  |

**Figure 4.** (**a**) Example of the reference vector $\mathbf{\Phi}$, and several instances of $\boldsymbol{\phi}$. (**b**) Empirical cdf and anomaly decision threshold $th_\mathbf{\Phi}$.

### 4.7. On the Implementation

Each network node $v_i \in \mathcal{V}_\tau$, must run online the $LC - IDS_{ij}$ for each neighboring node $v_j \in \mathcal{N}_i$; and each $LC - IDS_{ij}$ is designed by a supervised learning approach, which consists of a training stage and the online detection.

#### 4.7.1. Training Stage

During the training stage, we determine the decision threshold $th_g$ and the optimal values for the parameters $r_g$ and $\eta_g$, used to detect the $g$-th routing attack $\omega_g \in \Omega_\mathcal{A}$; as well as the anomaly decision threshold $th_\mathbf{\Phi}$. The value of each parameter $\theta_g$, can be chosen arbitrarily; all the parameters $\theta_g$ must be different among each other, because each $\theta_g$ defines a branch of the 'attack-constellation'. The decision thresholds $th_g$ and $th_\mathbf{\Phi}$, and the optimal parameters $r_g$ and $\eta_g$ are obtained from a set of training data $\{z_{\mathcal{A}g}, z_{\mathcal{N}g}\}$, which contains a set of attack poles $z_{\mathcal{A}g}$ labeled as $\mathcal{C}_{\mathcal{A}g}$, and a set of non-attack poles $z_{\mathcal{N}g}$, labeled as $\mathcal{C}_{\mathcal{N}g}$.

The set of label data $z_{\mathcal{A}g}$ is obtained by the $i$-th node $v_i \in \mathcal{V}_\tau$, by collecting input and output signals at a time when there is an $\omega_g$ attack present in the network. For the misuse-detection part of $LC - IDS_{ij}$, we use those measurements grouped in $d$ delayed samples, we obtain the system parameters of the multivariate linear regression model (8), $\alpha_b$ and $\beta_c$, $\forall b$ and $\forall c$. Those parameters are valid for a time-window that starts at $k - d$ and ends at $k$. We can define some search regions for the parameters $r_g$ and $\eta_g$. After that, we take a value of that search region $(r_g, \eta_g)$, to define the system input signals, $u_{\mathcal{A}_b}(k)$ and $u_{\mathcal{N}_c}(k)$, and the output signal $y_a(k)$. Then, we find the pole clusters, $|z_{\mathcal{A}g}|$, $|z_{\mathcal{N}g}|$, the decision threshold $th_g$ and the probability of error $P_g(\epsilon) = f_g(r_g, \eta_g)$ for those particular values of $r_g$ and $\eta_g$. We repeat this process for all the pair values of values $(r_g, \eta_g)$ to obtain the probability of error as a function $P_g(\epsilon) = f_g(r_g, \eta_g)$. Finally, we can solve the optimization problem in (26) to find the optimal parameters $r_g$, $\eta_g$ and their respective $th_g$ and $P(\epsilon)$.

To obtain the anomaly-detection threshold $th_\mathbf{\Phi}$, we use the non-attack data $\{z_{\mathcal{N}g} : g = 1, ..., M\}$ to obtain the vector $\mathbf{\Phi}$. This vector $\mathbf{\Phi}$, is a point of reference to characterize non-anomalous neighboring nodes. Then, we compare each instance of the vector $\boldsymbol{\phi}$ with the reference vector $\mathbf{\Phi}$. It is worth mentioning that the mean and standard deviation values that compose each instance of the vector $\boldsymbol{\phi}$, are obtained in a sliding-time-window fashion from the non-anomalous training data $\{z_{\mathcal{N}g} : g = 1, ..., M\}$. Those statistic parameters obtained from the time-window that starts at the time period $k - d_\mathbf{\Phi}$, and ends at the $k$-th period. With each instance of $\boldsymbol{\phi}$ and the reference vector $\mathbf{\Phi}$, we can obtain a set of

distance metrics $s_{\mathbf{\Phi}}$, and their respective cdf $F_{s_{\mathbf{\Phi}}}(s_{\mathbf{\Phi}})$, to finally obtain the anomaly decision threshold $th_{\mathbf{\Phi}}$.

Figure 5, shows the training process to find the decision thresholds $th_g$, $th_{\mathbf{\Phi}}$ and the optimal parameters $r_g$ and $\eta_g$.

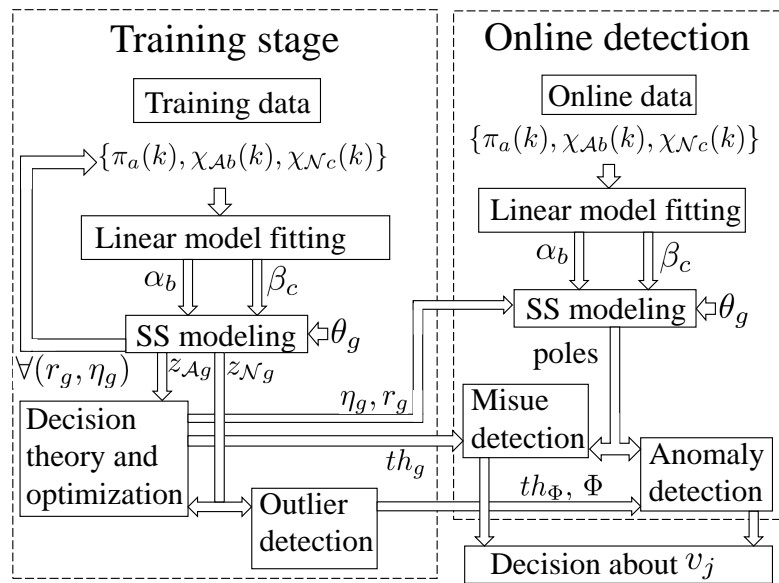

**Figure 5.** Training and online detection stages of $LC - IDS_{ij}$, during the training stage we obtain the state-space (SS) model and the respective optimal parameters and the optimal parameters $r_g$, $\eta_g$, $th_g$, $th_{\mathbf{\Phi}}$, $\mathbf{\Phi}$ that minimize the classification error probability. For online detection we obtain and compare the instantaneous system poles to the optimal threshold value, $th_g$, to detect each routing attack; and we use the reference vector $\mathbf{\Phi}$, and the threshold $th_{\mathbf{\Phi}}$, to detect anomalies.

### 4.7.2. Online Misuse and Anomaly Detection

Figure 5, describes online attack-detection process of $LC - IDS_{ij}$ to identify each $g$-th routing attack, or anomalous dynamic behavior of each $j$-th neighboring node $v_j \in \mathcal{N}_i$. The online attack-detection starts with the misuse decision threshold $th_g$, the anomaly decision threshold $th_{\mathbf{\Phi}}$, and the optimal parameters $r_g$ and $\eta_g$, obtained during the training stage. We form the input signals $u_{\mathcal{A}b}(k)$ and $u_{\mathcal{N}c}(k)$, and the output signal $y_a(k)$ from the multivariate linear regression model parameters and variables in (8). Then, we obtain the system poles of the 'attack-constellation' for that sampling period, and we compare the modulus of those poles to the decision threshold $th_g$ to decide if the neighboring node $v_j \in \mathcal{N}_i$ is an attacker. To find anomalous neighboring nodes, we compare the current value of the vector $\boldsymbol{\phi}$, to the reference vector $\mathbf{\Phi}$. Then, we obtain the current distance $s_{\mathbf{\Phi}}$, and compare it with the decision threshold $th_{\mathbf{\Phi}}$. Once an attacker has been identified, it can be added to a blacklist and the network administrator will receive an alert of the event. Please note that the data collection and the computation for intrusion detection is performed locally and individually by each network node to save network resources; however, by adding the attacker to a blacklist, each individual action causes a global impact as the attacker node is isolated from the network.

### 4.7.3. On the Computational Workload

In this subsection, we discuss on the computational workload required to implement the proposed technique, $LC - IDS_{ij}$, on a network node, to identify attackers and anomalous behaviors online, for each neighboring node $v_j \in \mathcal{N}_i$.

For each misuse detector in $LC - IDS_{ij}$, there exists a branch in the 'attack-constellation', and a modeling process that starts with a multivariate regression model. The number $n$, of parameters of the multivariate regression model in (8) equals the number of input signals considered. Please note that $n = \lambda_{ag} + \lambda_{ng} + 1 < A + N + 1$, because $|X_{\mathcal{A}g}| = \lambda_{ag} <$

$|X_{\mathcal{A}}| = A$; $|X_{\mathcal{N}_g}| = \lambda_{ng} < |X_{\mathcal{N}}| = N$. For each attack detector considered, $LC - IDS_{ij}$ has to estimate the parameters $\alpha_b$, $\beta_c$ and $\gamma_g$, by the least squares method, given the previous $d$, delayed measurements of the time series, $\pi_a(k)$, $\chi_{\mathcal{A}b}(k)$ and $\chi_{\mathcal{N}c}(k)$. The least squares method requires three matrix multiplications and an inverse matrix operation. It begins by multiplying two matrices, with dimensions $n \times d$ and $d \times n$, respectively, to obtain an $n \times n$ matrix. Then, we need to perform the most expensive operation in the least squares method that consists of calculating the inverse of the obtained matrix, whose dimensions are $n \times n$. The result must be multiplied for a matrix, whose dimensions are $n \times d$. Then, the obtained matrix has dimensions $n \times d$, and must be multiplied by a $d \times 1$ vector, to obtain the model parameters in a vector of dimensions $n \times 1$. The same operation must be repeated for each known routing attack detector in $LC - IDS_{ij}$.

Recall that the characteristic equation from which the model in $LC - IDS_{ij}$ originates, $Q(z)$ is of order $2M$ by definition, and it is composed of a total number $M$ of second order polynomials, $Q_g(z) : g = 1, ..., M$. To find the system poles of the 'attack-constellation', we need to find the roots of each second-degree polynomial $Q_g(z)$, which have a closed solution. This implies a potential computational cost reduction when compared to the calculation of the eigenvalues of the characteristic equation in (22).

The misuse-detection component of $LC - IDS_{ij}$, uses a decision rule $h_g(|z|)$ to make a decision about the $j$-th neighboring node. This decision rule compares the estimated poles of the 'attack-constellation' to the corresponding decision threshold $th_g$.

For the anomaly-detection engine of $LC - IDS_{ij}$, we need to calculate the mean values $\mu_{g,d_{\boldsymbol{\Phi}}}$, and standard deviations $\sigma_{g,d_{\boldsymbol{\Phi}}}$, of the moduli of the 'attack-constellation' poles, considering the current and previous $d_{\boldsymbol{\Phi}} - 1$ calculated poles. Then, we obtain the current distance metric $s_{\boldsymbol{\Phi}}$, between two vectors $\boldsymbol{\Phi} \in \mathcal{R}^{2M}$ and $\boldsymbol{\phi} \in \mathcal{R}^{2M}$, and compare it to the anomaly decision threshold $th_{\boldsymbol{\Phi}}$.

From the previous analysis, we can conclude that as the number of known routing attacks increases, the computational cost of $LC - IDSij$ increases as well. In order to keep the computational workload required by $LC - IDS_{ij}$ at a minimum, small models, with a small number of parameters, are desirable.

4.7.4. On the Time-to-Attack Detection

Please note that the time required by $LC - IDS_{ij}$ to identify an attack $\omega_g \in \Omega_{\mathcal{A}}$ launched by the $j$-th neighboring node, depends on the time-window length $d$, used to estimate the model parameters. The multivariate linear regression model in (8) has a total of $n = \lambda_a + \lambda_n + 1$ parameters; $d \geq n$. By increasing the number of input signal used for the parameter estimation, we may improve the attack-detection performance, but at the same time, the necessary time to detect the attack $\omega_g$ increases. Similarly, the time to detect anomalies, depends on the length of the time-window $d_{\boldsymbol{\Phi}}$. A larger length of $d_{\boldsymbol{\Phi}}$ reduces the number of false alarms, at the expense of increasing the time required to detect anomalies.

*4.8. On Unknown Attacks*

As previously stated, as new vulnerabilities are discovered in routing protocols, the number of known routing attacks would be continuously increasing. The anomaly-detection capabilities of $LC - IDS_{ij}$ can help us detect these new vulnerabilities, to then design the proper attack detectors, and include them into the 'attack-constellation'. Please note that because the system poles of the model in each $LC - IDS_{ij}$ move on predefined trajectories, they do not interfere with each other. Thus, we can repeat the training stage described in Section 4.7 to train as many new branches as necessary and include them to the 'attack-constellation', without affecting the attack-detection performance of the previously designed misuse detectors. The main drawback of this approach is the increasing complexity of the required computational resources of the technique. Collaborative approaches, in which different nodes detect a given subset of known attacks, might help

mitigate the increasing complexity problem of the technique, and are a possible future research direction.

## 5. Study Cases

To test the attack-detection performance of the proposed technique, a series of simulations were performed on an in-house developed event driven simulator, described in [15]. It is worth mentioning that to present a fair comparison of the proposed method with the prior [16], we are replicating the simulations presented in [16], comparing the misuse-detection results and obtaining anomaly-detection results of LC-IDS.

### 5.1. Simulation Parameters

We performed a series of 56 simulations for a wide variety of network scenarios. The simulation parameters are summarized in Table 2. The simulation period used is $T = 0.05$ s. The total simulated time was 20 s per each simulation. Each simulated scenario contains one attacker node, which launched the corresponding routing attack after the first 10 s of attack-free simulation. Four routing attacks were considered in the experiments, $\omega_1$ = route request flooding (RREQF), $\omega_2$ = selective forwarding (SF), $\omega_3$ = black hole (BH) and $\omega_4$ = worm hole (WH); $\Omega_{\mathcal{A}} = \{\omega_1, \omega_2, \omega_3, \omega_4\}$.

The simulations are divided in three experiments, the Node Density Experiment, the Attack Severity Experiment, and the Mobility Experiment. For the first experiment, we study the effects of node density and the position of the attacker on the attack-detection and anomaly-detection performance. We simulated the nodes at a random fixed position. The attack severity was fixed for all the attacks, $\psi_g = 0.1$. The total number of nodes in the scenario increased from the set, $\{65, 75, 85\}$. To assess the attacker's location impact, we repeat those experiments varying the attacker node position for each node density, first, the malicious node was set at the center of the scenario, then, it was set at the edge of the scenario. For the second experiment, we analyze the effects of different attack severity values on the misuse and anomaly-detection performance for each routing attack $\omega_g \in \Omega_{\mathcal{A}}$. No mobility was considered for this experiment. The attack severity was modified from the set $\{0, 0.1, 0.3, 0.5, 0.7\}$. For the RREQ flooding attack, the attack severity, $\psi_g$, was defined as the bandwidth consumption by the RREQ messages, normalized by the maximum channel capacity of the attacker's links. For the selective forwarding, black hole and worm hole attacks, the attack severity was defined as the probability of the attacker discarding data packets. For each routing attack, $\omega_g$, the attack severity, $\psi_g \in [0, 1)$. The third experiment compares the effects of mobility on the attack-detection and anomaly-detection performance, attack severity is fixed for all the attacks, $\psi_g = 0.1$, the number of simulated nodes is also fixed at 65. The mobility model used for this experiment is the random way point model, where each node speed is limited by a maximum speed value from the set, $\{2, 3, 4, 5\}$ m/s.

**Table 2.** Simulation parameters.

| Simulation Parameter | Parameter Value |
| --- | --- |
| Type of RWN | Ad hoc/Mobile ad hoc |
| Scenario dimensions | $80 \times 80$ m. |
| Total duration | 20 s. |
| Simulation period, $T$ | 0.05 s. |
| Number of nodes | 65/{65, 75, 85} |
| Mobility model | Static/Random Waypoint |
| Node speed | 0/{2, 3, 4, 5} m/s. |
| Number of attackers | 1 |
| Attack severity, $\psi_g$ | $\{0, 0.1\}/\{0, 0.1, 0.3, 0.5, 0.7\}$ |
| Node tx range | 15 m. |

**Table 2.** *Cont.*

| Simulation Parameter | Parameter Value |
|---|---|
| Floor noise | $-27$ dBm |
| Modulation | QPSK |
| MAC protocol | CSMA/CA |
| Routing protocol | AODV |
| Transport protocol | UDP |
| Traffic model | CBR |
| Type of attack | {RREQ Flooding, Selective Forwarding, Worm Hole, Black Hole} |

*5.2. Simulation Results*

Each attack detector $LC - IDS_{ij}$, is obtained by the methodology described in Section 4.7, to detect each routing attack $\omega_g \in \Omega_\mathcal{A}$, and to perform anomaly detection, for the three experiments.

The attack-detection performance of the misuse-detection component of each $LC - IDS_{ij}$, is evaluated in terms of detection accuracy ($DA_g$), the number of false positives ($FP_g$) and the number of false negatives ($FN_g$), for all the simulated scenarios. We are interested in testing the robustness of the proposed technique to a wide variety of network conditions, so that it could be implemented in low power devices. Thus, the time-to-attack-detection and computational requirements are minimal for all the analyzed scenarios. To achieve this minimization of computational resources, we consider the smallest possible detection model for each case. This smallest possible model consists of only one input signal and one output signal per routing attack; and those models are parameterized each time period using the minimum length possible for the misuse-detection time-window $d = 1$, and the minimum time-window for the anomaly detection $d_\Phi = 2$. The optimal parameters of the 'attack-constellation', $r_g$ and $\eta_g$, are presented for each case. Please note that by considering only one input signal per attack detector and a minimum length of the window size $d = 1$, the parameter estimation of the autoregressive model in (8) can be obtained by a simple division, eliminating the expensive operation of matrix inversion in the least squares approach proposed in Section 4.1. As mentioned in Section 4.7.3, we can obtain the roots of $M = 3$ second-degree polynomials by solving the general quadratic equation, which significantly reduces computation when compared to finding the roots of the characteristic polynomial in Equation (22).

To test the anomaly-detection performance of $LC - IDS_{ij}$, for each simulated scenario, we design an 'attack-constellation' that does not include the simulated routing attack in that particular scenario. For example, if the malicious node in one scenario launches the routing attack $\omega_3$, we design the attack-constellation to include the set of known attacks $\Omega_\mathcal{A} = \{\omega_1, \omega_2, \omega_4\}$. For each experiment we are considering a total of three known attacks. The fourth attack class is considered to be unknown to the IDS and is used to test the anomaly-detection performance of LC-IDS. Then we use the anomaly-detection accuracy ($DA_\Phi$), the number of false positive ($FP_\Phi$), and the number of false negatives ($FP_\Phi$), for that 'attack-constellation' and the previously unknown routing attack.

In Table 3, we show the input and output signals that were considered to obtain the multivariate linear regression parameters, for each $LC - IDS_{ij}$.

**Table 3.** Input and output signals used for each $LC - IDS_{ij}$ model.

| | RREQF | SF | BH | WH |
|---|---|---|---|---|
| $\pi_a(k)$ | received header bits | received bits per link | routing frequency of link | routing frequency of link |
| $\chi_{\mathcal{N}_c}(k)$ | $\varnothing$ | $\varnothing$ | $\varnothing$ | $\varnothing$ |
| $\chi_{\mathcal{A}_b}(k)$ | total received bits | bits sent per link | received packets | received packets |

The network metrics used to define the input and output signals of the LSI system, constitute the time series $\pi_a(k)$, $\chi_{\mathcal{A}_b}(k)$ and $\chi_{\mathcal{N}_c}(k)$ in Table 3, and are defined as,

- The received header bits, is a metric that measures the number of packet header bits received from the neighboring node's link to the attacker node at each simulation period, *k*.
- The total received bits, refers to the total number of received bits during one simulation period.
- The received bits per link, sent bits per link and received packets focus on the link of interest between the neighboring node and the attacker.
- The routing frequency of the link, measures the number of times that the link of interest appears as next hop in the routing tables normalized by the number of active routes.

To reduce the time-window length *d* used to obtain the model parameters, we include a pre-processing of the input and output signals before the adaptive fitting of the linear models. In this pre-processing stage, we filter the signals by a Butterworth low-pass filter. The main function of the low-pass filter, is to smooth the signals and to improve the signal-to-noise ratio. The Butterworth filter was designed as an analog low-pass filter with a cut off frequency $\omega_c = 0.24$ Hz and then it was converted to its digital form by the Tustin's bilinear transform with a sampling period $T = 0.05$ s. The transfer function in the $\mathcal{Z}$-plane, of the digital low-pass filter is,

$$H_{Bttr}(z) = \frac{0.000346z^2 + 0.00069217z + 0.000346}{z^2 - 1.947z + 0.9481}. \tag{33}$$

The second order filter was used to smoothen all the input and output signals for each simulated scenario. Please note that the misuse-detection evaluation parameters presented in Tables 4–7 are similar to those presented in [16].

5.2.1. Node Density Experiment

In Table 4, we present the attack-detection performance for the node density experiments, in which the attack node was placed at the center of the scenario. In general, we achieve good attack-detection performance for all the simulated scenarios, and for all the known attacks. The worst attack-detection performance ($DA = 97.346\%$) was obtained for the $\omega_2$ attack, for the 85-node scenario; the rest of misuse-detection accuracy results are >99.000%. Please note that this misuse-detection performance results were obtained considering only one input signal per known attack, and a minimum length of the time-window $d = 1$. This implies that the model parameters, $\alpha_b$, can be found by a single floating-point operation (a division), for each attack detector in $LC - IDS_{ij}$, every simulation period, $T$. Thus, in case it is necessary, we could improve the misuse-detection performance by considering more input signals or by increasing the time-window length $d$, at the expense of a higher computational workload. The optimal 'attack-constellation' parameters $r_g$, $\eta_g$ and $th_g$, were different for each simulated scenario, which implies that if the network conditions change significantly, it is necessary to obtain new optimal parameters for the 'attack-constellation'.

The anomaly-detection performance results are shown at the bottom of the misuse-detection results, for each simulated scenario. Most of the anomaly-detection accuracy results are good ($DA > 99.000\%$), even for the small time-window length $d_\Phi$. Please note that for all the anomaly-detection results, the number of false positives $FP_\Phi < 0.001\%$, and the number of false negatives $FN_\Phi$ contribute to most of the anomaly-detection error. The worst anomaly-detection accuracy ($DA_\Phi = 50.001\%$), was obtained for the case of 85 simulated node and an $\omega_2$ attack. This implies that in the presence of an $\omega_2$ attack, the attack-constellation composed of three branches, corresponding to $\omega_1$, $\omega_3$ and $\omega_4$; the anomaly detection component of $LC - IDS_{ij}$ will detect the anomalous behavior of the $j$-th neighboring node roughly one of every two sampling periods. This alert triggering rate might be sufficient to be noticed.

**Table 4.** Results for the number of nodes in the experiment. $DA_g$, $FP_g$, and $FN_g$ are given as a percentage.

| Nodes | | $\omega_1$ | $\omega_2$ | $\omega_3$ | $\omega_4$ |
|---|---|---|---|---|---|
| 65 | $r_g$ | 0.1 | 0.3 | 0.1 | 0.1 |
| | $\eta_g$ | 1.2 | 1.9 | 0.2 | 2.1 |
| | $DA_g$ | >99.999 | >99.999 | >99.999 | >99.999 |
| | $FP_g$ | <0.001 | <0.001 | <0.001 | <0.001 |
| | $FN_g$ | <0.001 | <0.001 | <0.001 | <0.001 |
| | $th_g$ | 0.0524 | 0.0930 | 0.0322 | 0.0605 |
| | $DA_\Phi$ | 79.496 | >99.999 | >99.999 | >99.999 |
| | $FP_\Phi$ | 0.001 | <0.001 | <0.001 | <0.001 |
| | $FN_\Phi$ | 20.503 | <0.001 | <0.001 | <0.001 |
| | $th_\Phi$ | 0.0173 | 0.0080 | 0.1190 | 0.0075 |
| 75 | $r_g$ | 0.1 | 0.6 | 0.1 | 0.1 |
| | $\eta_g$ | 2.3 | 35 | 0.2 | 2.5 |
| | $DA_g$ | 99.440 | >99.999 | >99.999 | >99.999 |
| | $FP_g$ | 0.137 | <0.001 | <0.001 | <0.001 |
| | $FN_g$ | 0.423 | <0.001 | <0.001 | <0.001 |
| | $th_g$ | 0.0518 | 0.4795 | 0.0735 | 0.0936 |
| | $DA_\Phi$ | >99.999 | >99.999 | 89.859 | 99.960 |
| | $FP_\Phi$ | <0.001 | <0.001 | 0.001 | <0.001 |
| | $FN_\Phi$ | <0.001 | <0.001 | 10.140 | 0.039 |
| | $th_\Phi$ | 0.0047 | 0.0036 | 0.0069 | 0.0073 |
| 85 | $r_g$ | 0.1 | 2.7 | 0.1 | 0.1 |
| | $\eta_g$ | 3.8 | 7.4 | 0.1 | 34.8 |
| | $DA_g$ | 99.974 | 97.346 | >99.999 | >99.999 |
| | $FP_g$ | 0.003 | 2.302 | <0.001 | <0.001 |
| | $FN_g$ | 0.023 | 0.352 | <0.001 | <0.001 |
| | $th_g$ | 0.0516 | 0.0645 | 0.0485 | 0.1 |
| | $DA_\Phi$ | >99.999 | 50.001 | >99.999 | 94.990 |
| | $FP_\Phi$ | <0.001 | <0.001 | <0.001 | 0.001 |
| | $FN_\Phi$ | <0.001 | 49.998 | <0.001 | 5.009 |
| | $th_\Phi$ | 0.4032 | 0.0246 | 0.0026 | 0.0066 |

In Table 5 we show the results for the simulated scenarios, in which we placed the attacker node at the edge of the scenario. Please note that we obtain better attack-detection performance for the cases in which the attacker node is placed at the center of the scenario than for the cases in which the attacker was placed at the edge of the scenario. This is due to the fact that when the attacker is at the center of the scenario, it has a larger number of neighboring nodes; thus, the attack has a larger impact on network performance. A larger impact on network performance, implies that it is easier to detect the routing attack.

**Table 5.** Results for the number of nodes in the experiment (Attacker at the edge of scenario, indicated by *). $DA_g$, $FP_g$, and $FN_g$ are given as a percentage.

| Nodes | | $\omega_1$ | $\omega_2$ | $\omega_3$ | $\omega_4$ |
|---|---|---|---|---|---|
| 65 * | $r_g$ | 0.5 | 0.1 | 20.7 | 0.1 |
| | $\eta_g$ | 7.6 | 0.3 | 1.5 | 4.9 |
| | $DA_g$ | 97.242 | 94.653 | >99.999 | >99.999 |
| | $FP_g$ | 0.966 | 2.811 | <0.001 | <0.001 |
| | $FN_g$ | 1.792 | 2.536 | <0.001 | <0.001 |
| | $th_g$ | 0.0637 | 0.0527 | 0.0391 | 0.0948 |
| | $DA_\Phi$ | 84.335 | 50.493 | >99.999 | >99.999 |
| | $FP_\Phi$ | <0.001 | <0.001 | <0.001 | <0.001 |
| | $FN_\Phi$ | 15.664 | 49.506 | <0.001 | <0.001 |
| | $th_\Phi$ | 0.0036 | 0.0311 | 0.0110 | 0.0033 |
| 75 * | $r_g$ | 0.1 | 0.1 | 0.1 | 0.1 |
| | $\eta_g$ | 1.1 | 0.6 | 0.2 | 23.7 |
| | $DA_g$ | >99.999 | 99.955 | >99.999 | >99.999 |
| | $FP_g$ | <0.001 | 0.023 | <0.001 | <0.001 |
| | $FN_g$ | <0.001 | 0.022 | <0.001 | <0.001 |
| | $th_g$ | 0.0544 | 0.0391 | 0.0348 | 0.0073 |
| | $DA_\Phi$ | >99.999 | >99.999 | >99.999 | 50.237 |
| | $FP_\Phi$ | <0.001 | <0.001 | <0.001 | <0.001 |
| | $FN_\Phi$ | <0.001 | <0.001 | <0.001 | 49.762 |
| | $th_\Phi$ | 0.0760 | 0.0034 | 0.0081 | 0.0184 |
| 85 * | $r_g$ | 16.3 | 0.1 | 9.5 | 0.6 |
| | $\eta_g$ | 35 | 9.3 | 1.3 | 35 |
| | $DA_g$ | 85.006 | 99.969 | >99.999 | >99.999 |
| | $FP_g$ | 0.654 | 0.028 | <0.001 | <0.001 |
| | $FN_g$ | 14.340 | 0.003 | <0.001 | <0.001 |
| | $th_g$ | 0.0027 | 0.1005 | 0.0083 | 0.600 |
| | $DA_\Phi$ | 80.667 | >99.999 | >99.999 | >99.999 |
| | $FP_\Phi$ | <0.001 | <0.001 | <0.001 | <0.001 |
| | $FN_\Phi$ | 19.332 | <0.001 | <0.001 | <0.001 |
| | $th_\Phi$ | 0.0472 | 6.0319 | 0.0013 | 0.0074 |

### 5.2.2. Attack Severity Experiment

Table 6, summarizes the misuse-detection and anomaly-detection performance for the different attack severity experiments. Please note that we achieve better results, compared to the node density experiments. Most of the $DA > 99.000\%$ for most of the simulated scenarios. The worst detection accuracy ($DA = 88.162\%$) was obtained for the $\omega_4$ and $\psi_3 = 0.3$ scenario. This is because, the greater attack severity values $\psi_g$, are associated with a higher impact on network performance degradation, making easier for the misuse-detection engine of $LC - IDS_{ij}$ to identify those attacks. As with the previous case, the time-window length for the adaptive fitting of the model parameters was $d = 1$. Thus, the model parameters, $\alpha_b$, can be found by a division for each attack-detection model $IDS_{ij,\omega_g}$, every simulation period, $kT$. This implies a minimum computational workload of $LC - IDS_{ij}$, and a minimum time-to-misuse-detection-time. Better misuse-detection results could be obtained by increasing the time-window length $d$, or by considering more input signals into the dynamical model of $LC - IDS_{ij}$. The optimal 'attack-constellation' parameters $r_g$, $\eta_g$ and $th_g$, are different for each simulated scenario.

As with the number of nodes experiment, the worst anomaly-detection results ($DA_\Phi = 50.001\%$), will produce a triggering alarm rate sufficient to be noticed. In addition, the majority of the anomaly-detection error is produced by the number of false negatives $FN_\Phi$, i.e., anomalous neighboring nodes detected as non-anomalous. The number of false positives is minimum, because of the way that the anomaly decision threshold $th_\Phi$ is defined.

**Table 6.** Results for the attack severity ($\psi_g$) experiment. $DA_g$, $FP_g$, and $FN_g$ are given as a percentage.

| $\psi_g$ | | $\omega_1$ | $\omega_2$ | $\omega_3$ | $\omega_4$ |
|---|---|---|---|---|---|
| | $r_g$ | 0.7 | 0.1 | 12.9 | 0.1 |
| | $\eta_g$ | 4.2 | 35 | 4.2 | 2.1 |
| | $DA_g$ | >99.999 | >99.999 | 99.930 | >99.999 |
| | $FP_g$ | <0.001 | <0.001 | 0.024 | <0.001 |
| | $FN_g$ | <0.001 | <0.001 | 0.046 | <0.001 |
| 10 | $th_g$ | 0.1084 | 0.1056 | $3.6 \times 10^{-9}$ | 0.0605 |
| | $DA_\Phi$ | 77.641 | >99.999 | 99.947 | >99.999 |
| | $FP_\Phi$ | <0.001 | <0.001 | <0.001 | <0.001 |
| | $FN_\Phi$ | 22.328 | <0.001 | 0.052 | <0.001 |
| | $th_\Phi$ | 0.0179 | 0.0067 | 0.0376 | 0.0156 |
| | $r_g$ | 0.7 | 0.1 | 0.1 | 0.1 |
| | $\eta_g$ | 3.8 | 35 | 0.1 | 0.1 |
| | $DA_g$ | >99.999 | >99.999 | >99.999 | 88.162 |
| | $FP_g$ | <0.001 | <0.001 | <0.001 | 0.370 |
| | $FN_g$ | <0.001 | <0.001 | <0.001 | 11.468 |
| 30 | $th_g$ | 0.1492 | 0.1056 | 0.0498 | 0.0716 |
| | $DA_\Phi$ | 50.001 | 99.858 | 84.613 | >99.999 |
| | $FP_\Phi$ | <0.001 | <0.001 | <0.001 | <0.001 |
| | $FN_\Phi$ | 49.999 | 0.141 | 15.386 | <0.001 |
| | $th_\Phi$ | 0.0874 | 0.0059 | 0.0265 | 0.0069 |
| | $r_g$ | 0.6 | 0.5 | 0.1 | 0.1 |
| | $\eta_g$ | 2.7 | 1.6 | 0.1 | 9.4 |
| | $DA_g$ | >99.999 | >99.999 | >99.999 | >99.999 |
| | $FP_g$ | <0.001 | <0.001 | <0.001 | <0.001 |
| | $FN_g$ | <0.001 | <0.001 | <0.001 | <0.001 |
| 50 | $th_g$ | 0.1426 | 0.1316 | 0.0491 | 0.100 |
| | $DA_\Phi$ | 50.915 | 99.858 | >99.999 | >99.999 |
| | $FP_\Phi$ | <0.001 | <0.001 | <0.001 | <0.001 |
| | $FN_\Phi$ | 49.084 | 0.141 | <0.001 | <0.001 |
| | $th_\Phi$ | 0.0816 | 0.0105 | 0.012 | 0.0024 |
| | $r_g$ | 0.1 | 0.1 | 9.5 | 0.1 |
| | $\eta_g$ | 3.5 | 14.1 | 31.5 | 11 |
| | $DA_g$ | >99.999 | >99.999 | 88.990 | >99.999 |
| | $FP_g$ | <0.001 | <0.001 | 0.178 | <0.001 |
| | $FN_g$ | <0.001 | <0.001 | 10.832 | <0.001 |
| 70 | $th_g$ | 0.0305 | 0.1049 | $6 \times 10^{-67}$ | 0.1 |
| | $DA_\Phi$ | 97.900 | 99.983 | >99.999 | 99.08 |
| | $FP_\Phi$ | <0.001 | <0.001 | <0.001 | <0.001 |
| | $FN_\Phi$ | 1.999 | 0.016 | <0.001 | 0.917 |
| | $th_\Phi$ | 0.0002 | 0.0101 | 0.0064 | 0.06469 |

### 5.2.3. Mobility Experiment

In Table 7, we summarize the attack-detection performance for the different mobility simulations. Please note that the mobility experiment results are not as good as for the previous experiments. This is originated from the highly dynamic network topology, which resulted in high uncertainty and dispersion of the model parameters, $\alpha_b$. For the misuse-detection case, most of the $DA > 90\%$, the worst $DA_g = 80.539\%$ was obtained for $\omega_4$ and a node speed of 4 m/s. Most of the anomaly-detection results $DA_\Phi > 98.000\%$, with the worst case $DA_\Phi = 50.001\%$, for the $\omega_1$ and 5 m/s case. However, better attack-detection results could be achieved by increasing the time-window length, $d$, or by considering more input signals in the dynamical models of $LC - IDS_{ij}$. Please note that similarly to the previous experiments results, the optimal 'attack-constellation' parameters $r_g$, $\eta_g$ and $th_g$, are different for each simulated scenario.

**Table 7.** Results for the mobility experiment. $DA_g$, $FP_g$, and $FN_g$ are given as a percentage. The first column represents the maximum node speed in (m/s).

| (m/s) | | $\omega_1$ | $\omega_2$ | $\omega_3$ | $\omega_4$ |
|---|---|---|---|---|---|
| | $r_g$ | 0.1 | 21.7 | 0.1 | 0.3 |
| | $\eta_g$ | 2.1 | 4.1 | 0.1 | 35 |
| | $DA_g$ | 93.923 | 90.746 | >99.999 | >99.999 |
| | $FP_g$ | 5.128 | 0.349 | <0.001 | <0.001 |
| | $FN_g$ | 0.949 | 9.254 | <0.001 | <0.001 |
| 2 | $th_g$ | 0.0361 | 0.0234 | 0.0505 | 0.300 |
| | $DA_\Phi$ | 99.580 | 53.557 | >99.999 | >99.999 |
| | $FP_\Phi$ | <0.001 | <0.001 | <0.001 | <0.001 |
| | $FN_\Phi$ | 0.419 | 46.442 | <0.001 | <0.001 |
| | $th_\Phi$ | 0.0669 | 0.2404 | 0.0356 | 0.3388 |
| | $r_g$ | 0.1 | 0.1 | 0.1 | 0.1 |
| | $\eta_g$ | 2.2 | 1.3 | 0.1 | 3.8 |
| | $DA_g$ | 98.540 | 99.947 | 98.433 | >99.999 |
| | $FP_g$ | 0.285 | 0.016 | 0.431 | <0.001 |
| | $FN_g$ | 1.175 | 0.037 | 1.136 | <0.001 |
| 3 | $th_g$ | 0.051 | 0.0345 | 0.0491 | 0.0914 |
| | $DA_\Phi$ | >99.999 | 50.287 | 99.477 | >99.999 |
| | $FP_\Phi$ | <0.001 | <0.001 | <0.001 | <0.001 |
| | $FN_\Phi$ | <0.001 | 49.712 | 0.522 | <0.001 |
| | $th_\Phi$ | 0.0167 | 0.4856 | 0.0279 | 0.0026 |
| | $r_g$ | 0.7 | 0.1 | 0.1 | 0.1 |
| | $\eta_g$ | 6.9 | 1.5 | 0.1 | 0.8 |
| | $DA_g$ | 99.500 | >99.999 | 86.946 | 80.539 |
| | $FP_g$ | 0.257 | <0.001 | 11.465 | 17.993 |
| | $FN_g$ | 0.243 | <0.001 | 1.589 | 1.468 |
| 4 | $th_g$ | 0.0860 | 0.0324 | 0.0483 | 0.065 |
| | $DA_\Phi$ | >99.999 | 98.791 | 50.082 | 99.851 |
| | $FP_\Phi$ | <0.001 | <0.001 | <0.001 | <0.001 |
| | $FN_\Phi$ | <0.001 | 1.208 | 49.917 | 0.148 |
| | $th_\Phi$ | 0.0046 | 0.0147 | 0.1183 | 0.0459 |
| | $r_g$ | 0.3 | 0.1 | 0.5 | 2.9 |
| | $\eta_g$ | 5.6 | 1.4 | 0.4 | 16.2 |
| | $DA_g$ | 99.381 | 85.820 | 99.868 | >99.999 |
| | $FP_g$ | 0.197 | 1.122 | 0.049 | <0.001 |
| | $FN_g$ | 0.422 | 3.058 | 0.083 | <0.001 |
| 5 | $th_g$ | 0.0616 | 0.0119 | 0.0555 | 0.1901 |
| | $DA_\Phi$ | 50.001 | >99.999 | >99.999 | 99.973 |
| | $FP_\Phi$ | <0.001 | <0.001 | <0.001 | <0.001 |
| | $FN_\Phi$ | 49.998 | <0.001 | <0.001 | 0.026 |
| | $th_\Phi$ | 2.1646 | 2.1342 | 0.0287 | 0.0131 |

## 6. Conclusions and Future Work

In this work, we have developed a general mathematical framework based on the theory of dynamical systems, to identify routing attacks and anomalous behaviors from the local perspective of an individual node in RWN. We expand the main idea of the root locus-misuse-detection technique presented in recent literature. By this dynamical systems perspective, we take advantage of the causal and temporal dependencies in the network data used to identify routing attacks. This allows us to overcome some of the open challenges in the state of the art of IDS for RWN described in Section 2.3, which are listed as follows,

- By modeling the dynamic behavior of neighboring nodes as a piecewise LSI system, we can represent all the relevant information to identify routing attacks on a

two-dimensional feature space, the $\mathcal{Z}$-plane. This can be thought of as an intrinsic dimensionality-reduction capability of the proposed technique. This reduction in the number of feature space dimensions does not require any additional dimensionality-reduction techniques as could be the case of Principal Component Analysis (PCA) or an autoencoder.

- By obtaining the state-space model for each $LC - IDS_{ij}$, we can represent the system poles for all the attack detectors on the same feature space, $\mathcal{Z}$-plane. This allows us to derive the 'attack-constellation' concept, which we use to perform misuse and anomaly detection.

- We develop a framework in which we can consider as many neighboring nodes and routing attacks, as necessary. In the case of the appearance of an unknown routing attack, we can repeat the training stage described in Section 4.7 to design a new attack detector and add a new branch to the current 'attack-constellation', without affecting the detection performance of the already considered attack detector. This property makes LC-IDS a flexible and scalable technique.

- The proposed intrusion detection technique is robust to a wide range of network conditions and is capable of online attack-detection and anomaly-detection without imposing excessive computing overhead and without consuming any network bandwidth, as can be noted from the detection accuracy and time-to-attack-detection results, and from the fact that each $LC - IDS_{ij}$ uses local information obtained from received data packets and incoming links to detect malicious neighboring nodes.

Please note that the experimental evidence suggests that the proposed technique can overcome some of the open challenges of the alternative approaches to intrusion detection mentioned in Section 2.1. Due to the local data collection and computation, LC-IDS does not consume network bandwidth, unlike collaborative approaches. Because LC-IDS models the dynamical behavior of neighboring nodes as linear systems for a given instant, each neighboring node can be represented by a quasi-static pole distribution on the $\mathcal{Z}$-plane, independently of the number of input signals considered; this dimensionally reduced feature space in which attack/anomaly detection takes place simplifies the problem of dynamic probability distributions and decision thresholds of statistical approaches. This dimensionality-reduction property that arises naturally in LC-IDS also implies a simplification when compared to machine-learning approaches that make use of feature extraction and dimensionality-reduction techniques in addition to the classification approach required for intrusion detection. By the other hand, our approach cannot overcome some of the open challenges in the literature; in the case of a network scenario with many nodes and high node mobility, LC-IDS will need to consider more than one input signal per attack detector, increasing significantly the computational requirements of the technique, as can be noted from Section 4.7.3.

As future work, we could explore the idea of using control theory to not just identify malicious neighboring nodes, but to allow an intelligent controller to take action on the network. This controller could take advantage of the two-dimensional latent space obtained by each $LC - IDS_{ij}$ that represents the dynamic behavior of neighboring nodes to control individual nodes behavior and their respective impact on global network performance to adaptively optimize the global network performance parameters (e.g., throughput, end-to-end delay).

**Author Contributions:** Conceptualization, J.Z.-M., R.V.-H. and C.V.-R.; methodology, J.Z.-M. and R.V.-H.; software, J.Z.-M.; validation, J.Z.-M., R.V.-H., C.V.-R. and M.Z.; formal analysis, J.Z.-M.; investigation, J.Z.-M.; resources, C.V.-R.; data curation, J.Z.-M.; writing—original draft preparation, J.Z.-M.; writing—review and editing, R.V.-H., C.V.-R. and M.Z.; visualization, J.Z.-M.; supervision, C.V.-R.; project administration, C.V.-R.; funding acquisition, C.V.-R. All authors have read and agreed to the published version of the manuscript.

**Funding:** This work was supported in part by the SEP-CONACyT Research Projects under Grants 255387 and 256237, the School of Engineering and Sciences and the Telecommunications and Networks Focus Group at Tecnologico de Monterrey.

**Institutional Review Board Statement:** Not applicable.

**Informed Consent Statement:** Not applicable.

**Data Availability Statement:** Not applicable.

**Conflicts of Interest:** The authors declare no conflict of interest.

## Appendix A. Derivation of the State-Space Representation

In this appendix, we obtain the state-space representation of the LSI system that models the dynamic behavior of the *j*-th neighboring node, $LC - IDS_{ij}$. We start from the set of multivariate linear regression equations. There is one equation per known attack $\omega_g \in \Omega_{\mathcal{A}}$, $g = 1, 2, ..., M$. The set of multivariate linear regression equations is,

$$\pi_1(k) = \sum_{b=1}^{A} \alpha_b \chi_{\mathcal{A}_b}(k) 1_{X_{\mathcal{A}_1}}(\chi_{\mathcal{A}_b}) + \sum_{c=1}^{N} \beta_c \chi_{\mathcal{N}_c}(k) 1_{X_{\mathcal{N}_1}}(\chi_{\mathcal{N}_c}) + \gamma_1(k), \tag{A1}$$

$$\vdots$$

$$\pi_M(k) = \sum_{b=1}^{A} \alpha_b \chi_{\mathcal{A}_b}(k) 1_{X_{\mathcal{A}_M}}(\chi_{\mathcal{A}_b}) + \sum_{c=1}^{N} \beta_c \chi_{\mathcal{N}_c}(k) 1_{X_{\mathcal{N}_M}}(\chi_{\mathcal{N}_c}) + \gamma_M(k). \tag{A2}$$

By substituting each $u_{\mathcal{A}_b}(k)$, $u_{\mathcal{N}_c}(k)$ and $y_g(k)$, from Equations (13)–(15), we obtain,

$$
\begin{aligned}
y_1(k) = &\sum_{b=1}^{A} \alpha_b u_{\mathcal{A}_b}(k) 1_{X_{\mathcal{A}_1}}(\chi_{\mathcal{A}_b}) + \sum_{b=1}^{A} \alpha_b^{\eta_1} y_1(k) 1_{X_{\mathcal{A}_1}}(\chi_{\mathcal{A}_b}) \\
&+ 2r_1 \cos\theta_1 \sum_{b=1}^{A} \alpha_b^{\eta_1} y_1(k-1) 1_{X_{\mathcal{A}_1}}(\chi_{\mathcal{A}_b}) \\
&- r_1^2 \sum_{b=1}^{A} \alpha_b^{\eta_1} y_1(k-2) 1_{X_{\mathcal{A}_1}}(\chi_{\mathcal{A}_b}) + \sum_{c=1}^{N} \beta_c u_{\mathcal{N}_c}(k) 1_{X_{\mathcal{N}_1}}(\chi_{\mathcal{N}_c}),
\end{aligned}
\tag{A3}
$$

$$\vdots$$

$$
\begin{aligned}
y_M(k) = &\sum_{b=1}^{A} \alpha_b u_{\mathcal{A}_b}(k) 1_{X_{\mathcal{A}_M}}(\chi_{\mathcal{A}_b}) + \sum_{b=1}^{A} \alpha_b^{\eta_M} y_M(k) 1_{X_{\mathcal{A}_M}}(\chi_{\mathcal{A}_b}) \\
&+ 2r_M \cos\theta_M \sum_{b=1}^{A} \alpha_b^{\eta_M} y_M(k-1) 1_{X_{\mathcal{A}_M}}(\chi_{\mathcal{A}_b}) \\
&- r_M^2 \sum_{b=1}^{A} \alpha_b^{\eta_M} y_M(k-2) 1_{X_{\mathcal{A}_M}}(\chi_{\mathcal{A}_b}) + \sum_{c=1}^{N} \beta_c u_{\mathcal{N}_c}(k) 1_{X_{\mathcal{N}_M}}(\chi_{\mathcal{N}_c}).
\end{aligned}
\tag{A4}
$$

Applying the $\mathcal{Z}$-transform, and then solving for each $y_g(k)$, we obtain,

$$
\begin{aligned}
Y_1(z) = &z^{-1} Y_1(z) \frac{2r_1 \cos\theta_1 \sum_{b=1}^{A} \alpha_b^{\eta_1} 1_{X_{\mathcal{A}_1}}(\chi_{\mathcal{A}_b})}{1 + \sum_{b=1}^{A} \alpha_b^{\eta_1} 1_{X_{\mathcal{A}_1}}(\chi_{\mathcal{A}_b})} - z^{-2} Y_1(z) \frac{r_1^2 \sum_{b=1}^{A} \alpha_b^{\eta_1} 1_{X_{\mathcal{A}_1}}(\chi_{\mathcal{A}_b})}{1 + \sum_{b=1}^{A} \alpha_b^{\eta_1} 1_{X_{\mathcal{A}_1}}(\chi_{\mathcal{A}_b})} \\
&+ U_{\mathcal{A}b}(z) \frac{\sum_{b=1}^{A} \alpha_b 1_{X_{\mathcal{A}_1}}(\chi_{\mathcal{A}_b})}{1 + \sum_{b=1}^{A} \alpha_b^{\eta_1} 1_{X_{\mathcal{A}_1}}(\chi_{\mathcal{A}_b})} + U_{\mathcal{N}c}(z) \frac{\sum_{c=1}^{N} \beta_c 1_{X_{\mathcal{N}_1}}(\chi_{\mathcal{N}_c})}{1 + \sum_{b=1}^{A} \alpha_b^{\eta_1} 1_{X_{\mathcal{A}_1}}(\chi_{\mathcal{A}_b})},
\end{aligned}
\tag{A5}
$$

$$\vdots$$

$$Y_M(z) = z^{-1} Y_M(z) \frac{2r_M \cos\theta_M \sum_{b=1}^{A} \alpha_b^{\eta_M} 1_{X_{A_M}}(\chi_{A_b})}{1 + \sum_{b=1}^{A} \alpha_b^{\eta_M} 1_{X_{A_M}}(\chi_{A_b})} - z^{-2} Y_M(z) \frac{r_M^2 \sum_{b=1}^{A} \alpha_b^{\eta_M} 1_{X_{A_M}}(\chi_{A_b})}{1 + \sum_{b=1}^{A} \alpha_b^{\eta_M} 1_{X_{A_M}}(\chi_{A_b})}$$
$$+ U_{Ab}(z) \frac{\sum_{b=1}^{A} \alpha_b 1_{X_{A_M}}(\chi_{A_b})}{1 + \sum_{b=1}^{A} \alpha_b^{\eta_M} 1_{X_{A_M}}(\chi_{A_b})} + U_{Nc}(z) \frac{\sum_{c=1}^{N} \beta_c 1_{X_{N_M}}(\chi_{N_c})}{1 + \sum_{b=1}^{A} \alpha_b^{\eta_M} 1_{X_{A_M}}(\chi_{A_b})}. \tag{A6}$$

Regrouping terms,

$$Y_1(z) = z^{-1} \left\{ Y_1(z) \frac{2r_1 \cos\theta_1 \sum_{b=1}^{A} \alpha_b^{\eta_1} 1_{X_{A_1}}(\chi_{A_b})}{1 + \sum_{b=1}^{A} \alpha_b^{\eta_1} 1_{X_{A_1}}(\chi_{A_b})} + z^{-1} \left[ -Y_1(z) \frac{r_1^2 \sum_{b=1}^{A} \alpha_b^{\eta_1} 1_{X_{A_1}}(\chi_{A_b})}{1 + \sum_{b=1}^{A} \alpha_b^{\eta_1} 1_{X_{A_1}}(\chi_{A_b})} \right] \right\}$$
$$+ U_{Ab}(z) \frac{\sum_{b=1}^{A} \alpha_b 1_{X_{A_1}}(\chi_{A_b})}{1 + \sum_{b=1}^{A} \alpha_b^{\eta_1} 1_{X_{A_1}}(\chi_{A_b})} + U_{Nc}(z) \frac{\sum_{c=1}^{N} \beta_c 1_{X_{N_1}}(\chi_{N_c})}{1 + \sum_{b=1}^{A} \alpha_b^{\eta_1} 1_{X_{A_1}}(\chi_{A_b})}, \tag{A7}$$

$$\vdots$$

$$Y_M(z) = z^{-1} \left\{ Y_M(z) \frac{2r_M \cos\theta_M \sum_{b=1}^{A} \alpha_b^{\eta_M} 1_{X_{A_M}}(\chi_{A_b})}{1 + \sum_{b=1}^{A} \alpha_b^{\eta_M} 1_{X_{A_M}}(\chi_{A_b})} + z^{-1} \left[ -Y_M(z) \frac{r_M^2 \sum_{b=1}^{A} \alpha_b^{\eta_M} 1_{X_{A_M}}(\chi_{A_b})}{1 + \sum_{b=1}^{A} \alpha_b^{\eta_M} 1_{X_{A_M}}(\chi_{A_b})} \right] \right\}$$
$$+ U_{Ab}(z) \frac{\sum_{b=1}^{A} \alpha_b 1_{X_{A_M}}(\chi_{A_b})}{1 + \sum_{b=1}^{A} \alpha_b^{\eta_M} 1_{X_{A_M}}(\chi_{A_b})} + U_{Nc}(z) \frac{\sum_{c=1}^{N} \beta_c 1_{X_{N_M}}(\chi_{N_c})}{1 + \sum_{b=1}^{A} \alpha_b^{\eta_M} 1_{X_{A_M}}(\chi_{A_b})}. \tag{A8}$$

The state variable can be defined as,

$$X_1^{(1)}(z) = Y_1(z), \tag{A9}$$

$$X_1^{(2)}(z) = -z^{-1} X_1^{(1)}(z) \frac{r_1^2 \sum_{b=1}^{A} \alpha_b^{\eta_1} 1_{X_{A_1}}(\chi_{A_b})}{1 + \sum_{b=1}^{A} \alpha_b^{\eta_1} 1_{X_{A_1}}(\chi_{A_b})}, \tag{A10}$$

$$\vdots$$

$$X_M^{(1)}(z) = Y_M(z), \tag{A11}$$

$$X_M^{(2)}(z) = -z^{-1} X_M^{(1)}(z) \frac{r_M^2 \sum_{b=1}^{A} \alpha_b^{\eta_M} 1_{X_{A_M}}(\chi_{A_b})}{1 + \sum_{b=1}^{A} \alpha_b^{\eta_M} 1_{X_{A_M}}(\chi_{A_b})}. \tag{A12}$$

Thus, the state transition equations are,

$$zX_1^{(1)}(z) = X_1^{(1)}(z) \frac{2r_1 \cos\theta_1 \sum_{b=1}^{A} \alpha_b^{\eta_1} 1_{X_{A_1}}(\chi_{A_b})}{1 + \sum_{b=1}^{A} \alpha_b^{\eta_1} 1_{X_{A_1}}(\chi_{A_b})} + X_1^{(2)}, \tag{A13}$$

$$zX_1^{(2)}(z) = -X_1^{(1)}(z) \frac{r_1^2 \sum_{b=1}^{A} \alpha_b^{\eta_1} 1_{X_{A_1}}(\chi_{A_b})}{1 + \sum_{b=1}^{A} \alpha_b^{\eta_1} 1_{X_{A_1}}(\chi_{A_b})} + U_{Ab}(z) \frac{\sum_{b=1}^{A} \alpha_b 1_{X_{A_M}}(\chi_{A_b})}{1 + \sum_{b=1}^{A} \alpha_b^{\eta_M} 1_{X_{A_M}}(\chi_{A_b})}$$
$$+ U_{Nc}(z) \frac{\sum_{c=1}^{N} \beta_c 1_{X_{N_1}}(\chi_{N_c})}{1 + \sum_{b=1}^{A} \alpha_b^{\eta_1} 1_{X_{A_1}}(\chi_{A_b})}, \tag{A14}$$

$$\vdots$$

$$zX_M^{(1)}(z) = X_M^{(1)}(z) \frac{2r_M \cos\theta_M \sum_{b=1}^{A} \alpha_b^{\eta_M} 1_{X_{A_M}}(\chi_{A_b})}{1 + \sum_{b=1}^{A} \alpha_b^{\eta_M} 1_{X_{A_M}}(\chi_{A_b})} + X_M^{(2)}, \tag{A15}$$

$$zX_M^{(2)}(z) = -X_M^{(1)}(z)\frac{r_M^2 \sum_{b=1}^{A} \alpha_b^{\eta_M} 1_{X_{A_M}}(\chi_{A_b})}{1 + \sum_{b=1}^{A} \alpha_b^{\eta_M} 1_{X_{A_M}}(\chi_{A_b})} + U_{Ab}(z)\frac{\sum_{b=1}^{A} \alpha_b 1_{X_{A_M}}(\chi_{A_b})}{1 + \sum_{b=1}^{A} \alpha_b^{\eta_M} 1_{X_{A_M}}(\chi_{A_b})}$$
$$+ U_{\mathcal{N}c}(z)\frac{\sum_{c=1}^{N} \beta_c 1_{X_{\mathcal{N}_M}}(\chi_{\mathcal{N}_c})}{1 + \sum_{b=1}^{A} \alpha_b^{\eta_M} 1_{X_{A_M}}(\chi_{A_b})}. \tag{A16}$$

The output equations are,

$$Y_1(z) = X_1^{(1)}(z), \tag{A17}$$

$$\vdots$$

$$Y_M(z) = X_M^{(1)}(z). \tag{A18}$$

Applying the inverse $\mathcal{Z}$-transform, and arranging in matrix form the obtained state transition equations, we obtain the state equations in the time domain, as in Equation (16). Similarly, the output equations in the time domain and in matrix form are the same as the ones in Equation (20).

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
