# Peer review of "LC-IDS: Loci-Constellation-Based Intrusion Detection for Reconfigurable Wireless Networks"

_electronics, doi:10.3390/electronics10243053_

Round 1
Reviewer 1 Report
The authors propose an intrusion detection mechanism for reconfigurable wireless networks. The paper is well-structured and well-written; however, I think there are some major concerns in this current work.
- The attack detection mechanism is unclear. If each node has an IDS, which node gets the global information by collecting local information of all nodes? Does the attacker node have an IDS?
- Please explain how the distributed IDSes in each node cooperate to detect routing attacks.
- What is the overhead of the proposed intrusion detection mechanism? The authors mentioned in Section 2 that some ML-based NIDS have a high computational workload. Please verify the proposed mechanism has a low overhead compared to other studies.
- The experimental analysis is fairly weak. The authors need to verify and evaluate their proposed mechanism in various environments with other closely related works including their previous works [14][15].
- The authors discuss detecting unknown attacks in the paper; however, the entire framework is based on known vulnerabilities and attacks, hence it is not clear how the setup handles unknown attack cases in the experimental analysis.
Reviewer 2 Report
In this paper, the authors introduce Loci-Constellation-based Intrusion Detection System (LC-IDS), a generalized mathematical framework that designs an IDS capable of misuse and anomaly detection on a two-dimensional feature space, adapted for routing intrusion detection in Reconfigurable Wireless Networks (RWN).
The authors demonstrate their method in detail and this study is an innovation in the sense that it exploits mathematical modeling to represent an attack and/or an anomaly during a given time, on a Z-plane. The implementation is done on each node of the network and adapts to the mobility of the network (displacement, deletion, addition, deterioration of a node, ...). The innovation is nevertheless weighted in the sense that this study follows from one of their previous work on misuse detection (15) within this same type of network, which can be considered as an improvement.
The results and their interpretations seem mastered, and the projection on the future work consisting in making the network capable of correcting its flows in order to adapt to attacks/anomalies seems interesting.
However, a main drawback of the paper is that there is no concrete comparison of the results with the state of the art in detecting anomalies and attacks on RWN. There should at least be a comparison of the results on the misuse detection of method from article [15]. And there is not enough recent related work on IDS mentioned in the paper, despite a lot of research work in this domain.
The representation of concepts and results by diagrams and tables is good, despite a difference in readability between the methodology and experimentation sections. In order to improve the ease of reading and understanding, the theoretical and mathematical explanation of the method should be synthesized.
The quantitative data given in the experimentation section seem complete for third-party replication, but information on tools and technologies are missing.
In conclusion the paper is rather well written, despite some flaws on the form, and represents a real interest for the scientific community of the network and telecommunication sector, and in the case of a usable technical implementation of their method.
Here are some detailed remarks.
- It would be useful to introduce in a general way the notion of "Linear Shift-Invariant" before talking about its implementation.
- It would be useful to explain what an "online attack" is.
- In Figure 4 a, a little extra would be to demonstrate the decision boundary for the non-attack region.
- In Figure 5, could you please clarify what SS means
- In Figure 5, it should be « online » instead of « on line »
- In line 847 : It is mentioned that in Table 6 the worst detection accuracy for misuse case is 88.990% and it was obtained with attack ω3 and attack severity ψg of 0.7 But according to Table 6, the worst detection accuracy for misuse case is 88.162%, obtained with attack ω4 and attack severity ψg 0.3
- In line 859: It is mentioned that in Table 6 the worst detection accuracy for anomalies is 50.915% and it was obtained with attack ω1 and attack severity ψg 0.5. But according to Table 6, the worst detection accuracy for anomalies is 50.001%, obtained with attack ω1 and attack severity ψg 0.3
- Table 7: It would have been interesting to comment and explain the results (best and worst cases) as for Tables 4, 5 and 6.
- Error line 233: should be « constraints » instead of « constrains »
- Error line 275: should be « we » instead of « we we »
- Error line 434: in « We discuss the implementation and online attack and anomaly detection of LC-IDS » => the first « and » should be replaced by "of" ?
- Error line 480: should add « at » before « each sampling period »
- Error line 576: should be « greater » instead of « grater »
- Error Table 2: should remove space after « Black Hole » at last line
- Error line 859: there is a syntax error with the sentence "will produce a triggering alarm rate is sufficient to be noticed."
Reviewer 3 Report
Aimed at the detection of anomaly routing in reconfigurable wireless networks, a framework named Loci-Constellation-based featuring low computational complexity and distributed nature is proposed in this study.
The article is well organized with good linguistic quality.
This study has its contribution in that an elaborated math model for the proposed framework is developed.
Simulations demonstrate the feasibility of the proposed scheme.
Relative merit and disadvantage in comparison with existing approaches are not well addressed.
A comparative study will render this work more persuasive.
